# Black-Box Batch Active Learning for Regression

**Andreas Kirsch**                                                                  *andreas.kirsch@cs.ox.ac.uk*
*OATML, Department of Computer Science*
*University of Oxford*

**Reviewed on OpenReview:** *https://openreview.net/forum?id=fvEvDlKko6*

## Abstract

Batch active learning is a popular approach for efficiently training machine learning models on large, initially unlabelled datasets by repeatedly acquiring labels for batches of data points. However, many recent batch active learning methods are *white-box approaches* and are often limited to differentiable parametric models: they score unlabeled points using acquisition functions based on model embeddings or first- and second-order derivatives. In this paper, we propose *black-box batch active learning* for regression tasks as an extension of white-box approaches. Crucially, our method only relies on model predictions. This approach is compatible with a wide range of machine learning models including regular and Bayesian deep learning models and non-differentiable models such as random forests. It is rooted in Bayesian principles and utilizes recent kernel-based approaches. This allows us to extend a wide range of existing state-of-the-art white-box batch active learning methods (BADGE, BAIT, LCMD) to black-box models. We demonstrate the effectiveness of our approach through extensive experimental evaluations on regression datasets, achieving surprisingly strong performance compared to white-box approaches for deep learning models.

## 1 Introduction

By selectively acquiring labels for a subset of available unlabelled data, active learning (Cohn et al., 1994) is suited for situations where the acquisition of labels is costly or time-consuming, such as in medical imaging or natural language processing. However, in deep learning, many recent batch active learning methods have focused on *white-box approaches* that rely on the model being parametric and differentiable, using first or second-order derivatives (e.g. model embeddings)[1].

This can present a limitation in real-world scenarios where model internals or gradients might be expensive to access—or might not be accessible at all. This is particularly of concern for 'foundation models' (Bommasani et al., 2021) and large language models such as GPT-3 (Brown et al., 2020), for example, when accessed via a third-party API. More generally, a lack of differentiability might hinder application of white-box batch active learning approaches to non-differentiable models.

To address these limitations, we examine *black-box batch active learning (B³AL)*[2] for regression which is compatible with a wider range of machine learning models. By *black-box*, we mean that our approach only relies on model predictions and does not require access to model internals or gradients. Our approach is rooted in Bayesian principles and only requires model predictions from a small (bootstrapped) ensemble. Specifically, we utilize an *(empirical) predictive covariance kernel* based on sampled predictions. We show that the well-known gradient kernel (Kothawade et al., 2021; 2022; Ash et al., 2019; 2021) can be seen as an approximation of this predictive covariance kernel.

The proposed approach extends to non-differentiable models through a Bayesian view on the hypothesis space formulation of active learning, based on the ideas behind query-by-committee (Seung et al., 1992). This

---

[1]Model embeddings can also be seen as first-order derivatives of the score under regression in regard to the last layer.
[2]The code is available at https://github.com/BlackHC/2302.08981.

enables us to use batch active learning methods, such as BAIT (Ash et al., 2021) and BADGE (Ash et al., 2019) in a black-box setting with non-differentiable models, such as random forests or gradient-boosted trees.

By leveraging the strengths of kernel-based methods and Bayesian principles, our approach improves the labeling efficiency of a range of differentiable and non-differentiable machine-learning models with surprisingly strong performance. Specifically, we evaluate black-box batch active learning on a diverse set of regression datasets. We focus on regression because classification using the same approach would require Laplace approximations, Monte Carlo sampling, or expectation propagation (Williams & Rasmussen, 2006; Hernández-Lobato et al., 2011). This would complicate a fair comparison between black-box and white-box approaches. For the same reason, we also do not consider proxy-based active learning methods (Coleman et al., 2019), which constitutes an orthogonal direction to our investigation. Unlike the above mentioned *white-box* parametric active learning methods which scale in the number of (last-layer) model parameters or the embedding size, our method scales in the number of drawn predictions, and we find that we can already obtain excellent results with a small number of them. Our results demonstrate the label efficiency of B³AL for various machine learning models. For deep learning models, B³AL even performs better than the corresponding state-of-the-art white-box methods in many cases.

The rest of the paper is organized as follows: in §2, we discuss related work in active learning and kernel-based methods. In §3, we describe the relevant background and provide a detailed description of B³AL. In §4, we detail the experimental setup and provide the results of our experimental evaluation. Finally, §5 concludes with a discussion and directions for future research.

## 2 Related Work

Active learning has a rich history, with its origins dating back to seminal works such as Cohn et al. (1994); Lindley (1956); Fedorov (1972); MacKay (1992). A comprehensive survey of early active learning methods can be found in Settles (2009), while more recent surveys of contemporary deep learning methods can be found in Ren et al. (2021) and Zhan et al. (2022). Our work focuses on pool-based batch active learning, which involves utilizing a pool set of unlabeled data and acquiring labels for batches of points rather than individual points at a time. The main challenge in pool-based batch active learning is the choice of the acquisition function.

**Differentiable Models**. Many acquisition functions approximate well-known information-theoretic quantities (MacKay, 1992), often by approximating Fisher information implicitly or explicitly (Kirsch & Gal, 2022). This can be computationally expensive, particularly in deep learning where the number of model parameters can be large—even when using last-layer approximations or assuming a generalized linear model. BADGE (Ash et al., 2019) and BAIT (Ash et al., 2021) approximate the Fisher information using last-layer loss gradients or the Hessian, respectively, but still have a computational cost scaling with the number of last layer weights. This also applies to methods using similarity matrices (kernels) based on loss gradients of last-layer weights such as SIMILAR (Kothawade et al., 2022) and PRISM (Kothawade et al., 2022), to only name a few. Importantly, all of these approaches require differentiable models.

**Non-Differentiable Models.** *Query-by-committee (QbC, Seung et al. (1992))* measures the disagreement between different model instances to identify informative samples and has been applied to regression (Krogh & Vedelsby, 1994; Burbidge et al., 2007). Kee et al. (2018) extend QbC to the batch setting with a diversity term based on the distance of data points in input space. Nguyen et al. (2012) show batch active learning for random forests. They train an ensemble of random forests and evaluate the joint entropy of the predictions of the ensemble for batch active learning, which can be seen as a special case of BatchBALD in regression.

**BALD.** Our work most closely aligns with the BALD-family of Bayesian active learning acquisition functions (Houlsby et al., 2011), which focus on classification tasks, however. The crucial insight of BALD is applying the symmetry of mutual information to compute the expected information gain in prediction space instead of in parameter space. As a result, BALD is a *black-box technique* that only leverages model predictions. The extension of BALD to deep learning and multi-class classification tasks using Monte-Carlo dropout models is presented in Gal et al. (2017), which employs batch active learning by selecting the top-$B$ scoring samples from the pool set in each acquisition round. BatchBALD (Kirsch et al., 2019) builds upon BALD

by introducing a principled approach for batch active learning and consistent MC dropout to account for correlations between predictions. The estimators utilized by Gal et al. (2017) and Kirsch et al. (2019) enumerate over all classes, leading to a trade-off between combinatorial explosion and Monte-Carlo sampling, which can result in degraded quality estimates as acquisition batch sizes increase. Houlsby et al. (2011), Gal et al. (2017), and Kirsch et al. (2019) have not applied BALD to regression tasks. More recently, Jesson et al. (2021) investigate active learning for regressing causal treatment effects and adopt a sampling distribution based on individual-acquisition scores, a heuristic proposed in Kirsch et al. (2021).

**Kernel-Based Methods.** Holzmüller et al. (2022) examine the previously mentioned methods and unify them using gradient-based kernels. Specifically, they express BALD (Houlsby et al., 2011), BatchBALD (Kirsch et al., 2019), BAIT (Ash et al., 2021), BADGE (Ash et al., 2019), ACS-FW (Pinsler et al., 2019), and Core-Set (Sener & Savarese, 2017)/FF-Active (Geifman & El-Yaniv, 2017) using kernel-based methods for regression tasks. They also propose a new method, LCMD (largest cluster maximum distance).

**Our Contribution.** We extend the work of Houlsby et al. (2011) and Holzmüller et al. (2022) by combining the prediction-based approach with a kernel-based formulation. This trivally enables batch active learning on regression tasks using black-box predictions for a wide range of existing batch active learning methods.

## 3 Methodology

Here, we describe the problem setting and motivate the proposed method.

**Problem Setting.** Our proposed method is inspired by the BALD-family of active learning frameworks (Houlsby et al., 2011) and its extension to batch active learning (Kirsch et al., 2019). In our derviation, we make use of a Bayesian model in the narrow sense that we require some stochastic parameters $\boldsymbol{\Omega}$—the model parameters or bootstrapped training data in the case of non-differentiable models like random forests, for example—with a distribution[3] $p(\boldsymbol{\omega})$ (Bickford Smith et al., 2023):

$$p(y, \boldsymbol{\omega} \mid \mathbf{x}) = p(y \mid \mathbf{x}, \boldsymbol{\omega}) \, p(\boldsymbol{\omega}). \tag{1}$$

Bayesian model averaging (BMA) is performed by marginalizing over $p(\boldsymbol{\omega})$ to obtain the predictive distribution $p(y \mid \mathbf{x})$. We can use Bayesian inference to obtain a posterior $p(\boldsymbol{\omega} \mid \mathcal{D})$ for additional data $\mathcal{D}$ via:

$$p(\boldsymbol{\omega} \mid \mathcal{D}) \propto p(\mathcal{D} \mid \boldsymbol{\omega}) \, p(\boldsymbol{\omega}), \tag{2}$$

where $p(\mathcal{D} \mid \boldsymbol{\omega})$ is the likelihood of the data given the parameters $\boldsymbol{\Omega}$ and $p(\boldsymbol{\omega} \mid \mathcal{D})$ is the (new) posterior distribution over $\boldsymbol{\Omega}$. Importantly, the choice of $p(\boldsymbol{\omega})$ covers ensembles (Breiman, 1996; Lakshminarayanan et al., 2017) as well as models with additional stochastic inputs (Osband et al., 2021) or randomized training data by subsampling of the training set, e.g., bagging (Breiman, 1996).

**Pool-based Active Learning** assumes access to a pool set of unlabeled data $\mathcal{D}^{\text{pool}} = \{\mathbf{x}_i^{\text{pool}}\}$ and a small initially labeled training set $\mathcal{D}^{\text{train}} = \{(\mathbf{x}_i^{\text{train}}, y_i^{\text{train}})\}$, or $\mathcal{D}^{\text{train}} = \emptyset$. In the batch acquisition setting, we want to repeatedly acquire labels for a subset $\{\mathbf{x}_i^{\text{acq}}\}$ of the pool set of a given acquisition batch size $B$ and add them to the training set $\mathcal{D}^{\text{train}}$. Ideally, we want to select samples that are highly 'informative' for the model. For example, these could be samples that are likely to be misclassified or have a large prediction uncertainty for models trained on the currently available training set $\mathcal{D}^{\text{train}}$. Once we have chosen such an *acquisition batch* $\{\mathbf{x}_i^{\text{acq}}\}$ of unlabeled data, we acquire labels $\{y_i^{\text{acq}}\}$ for these samples and train a new model on the combined training set $\mathcal{D}^{\text{train}} \cup \{(\mathbf{x}^{\text{acq}}_i, y^{\text{acq}}_i)\}$ and repeat the process. Crucial to the success of active learning is the choice of acquisition function $\mathcal{A}(\{\mathbf{x}_i^{\text{acq}}\}; p(\boldsymbol{\omega}))$ which is a function of the acquisition batch $\{\mathbf{x}_i^{\text{acq}}\}$ and the distribution $p(\boldsymbol{\omega})$ and which we try to maximize in each acquisition round. It measures the informativeness of an acquisition batch for the current model.

**Univariate Regression** is a common task in machine learning. We assume that the target $y$ is real-valued ($\in \mathbb{R}$) with homoscedastic Gaussian noise:

$$Y \mid \mathbf{x}, \boldsymbol{\omega} \sim \mathcal{N}(\mu(\mathbf{x}; \boldsymbol{\omega}), \sigma_N^2). \tag{3}$$

---

[3]We do not require a prior distribution as active learning is not concerned with how we arrive at the model we want to acquire labels for. We can define a prior and perform, e.g, variational inference, but we do not *need* to. Hence, we use $p(\boldsymbol{\omega})$.

Equivalently, $Y \mid \mathbf{x}, \boldsymbol{\omega} \sim \mu(\mathbf{x}; \boldsymbol{\omega}) + \varepsilon$ with $\varepsilon \sim \mathcal{N}(0, \sigma_N^2)$. As usual, we assume that the noise is independent for different inputs $\mathbf{x}$ and parameters $\boldsymbol{\omega}$. Homoscedastic noise is a special case of the general heteroscedastic setting: the noise variance is simply a constant. Our approach can be extended to heteroscedastic noise by substituting a function $\sigma_N(\mathbf{x}; \boldsymbol{\omega})$ for $\sigma_N$, but for this work we limit ourselves to the simplest case.

### 3.1 Kernel-based Methods

We build on Holzmüller et al. (2022) which expresses contemporary batch active learning methods using kernel-based methods. While a full treatment is naturally beyond the scope of this paper, we briefly review some key ideas here. We refer the reader to the extensive paper for more details.

**Gaussian Processes** are one way to introduce kernel-based methods. A simple way to think about Gaussian Processes (Williams & Rasmussen, 2006; Lázaro-Gredilla & Figueiras-Vidal, 2010; Rudner et al., 2022) is as Bayesian linear regression model with an implicit, potentially infinite-dimensional feature space (depending on the covariance kernel) that uses the kernel trick to abstract away the feature map from input space to feature space.

**Multivariate Gaussian Distribution.** The distinctive property of a Gaussian Process is that all predictions are jointly Gaussian distributed. We can then write the joint distribution for a univariate regression model as:

$$Y_1, \ldots, Y_n \mid \mathbf{x}_1, \ldots, \mathbf{x}_n \sim \mathcal{N}(\mathbf{0}, \text{Cov}[\mu(\mathbf{x}_1), \ldots, \mu(\mathbf{x}_n)] + \sigma_N^2 \mathbf{I}), \tag{4}$$

where $\mu(\mathbf{x})$ are the observation-noise free predictions as random variables and $\text{Cov}[\mu(\mathbf{x}_1), \ldots, \mu(\mathbf{x}_n)]$ is the covariance matrix of the predictions. The covariance matrix is defined via the kernel function $k(\mathbf{x}, \mathbf{x}')$:

$$\text{Cov}[\mu(\mathbf{x}_1), \ldots, \mu(\mathbf{x}_n)] = \left[ k(\mathbf{x}_i, \mathbf{x}_j) \right]_{i,j=1}^{n,n}. \tag{5}$$

The kernel function $k(\mathbf{x}, \mathbf{x}')$ can be chosen almost arbitrarily, e.g. see Williams & Rasmussen (2006, Ch. 4). The linear kernel $k(\mathbf{x}, \mathbf{x}') = \langle \mathbf{x}, \mathbf{x}' \rangle$ and the radial basis function kernel $k(\mathbf{x}, \mathbf{x}') = \exp(-\frac{1}{2}|\mathbf{x} - \mathbf{x}'|^2)$ are common examples, as is the gradient kernel, which we examine next.

**Fisher Information & Linearization.** When using neural networks for regression, the gradient kernel

$$k_{\text{grad}}(\mathbf{x}; \mathbf{x}' \mid \boldsymbol{\omega}^*) \triangleq \nabla_{\boldsymbol{\omega}} \mu(\mathbf{x}; \boldsymbol{\omega}^*) \nabla_{\boldsymbol{\omega}}^2 [-\log p(\boldsymbol{\omega}^*)]^{-1} \nabla_{\boldsymbol{\omega}} \mu(\mathbf{x}'; \boldsymbol{\omega}^*)^\top \tag{6}$$

$$= \langle \nabla_{\boldsymbol{\omega}} \mu(\mathbf{x}; \boldsymbol{\omega}^*), \nabla_{\boldsymbol{\omega}} \mu(\mathbf{x}'; \boldsymbol{\omega}^*) \rangle_{\nabla_{\boldsymbol{\omega}}^2 [-\log p(\boldsymbol{\omega}^*)]^{-1}} \tag{7}$$

is the canonical choice, where $\boldsymbol{\omega}^*$ is a *maximum likelihood* or *maximum a posteriori estimate (MLE, MAP)* and $\nabla_{\boldsymbol{\omega}}^2 [-\log p(\boldsymbol{\omega}^*)]$ is the Hessian of the negative log likelihood at $\boldsymbol{\omega}^*$. Note that $\nabla_{\boldsymbol{\omega}} \mu(\mathbf{x}; \boldsymbol{\omega}^*)$ is a *row vector*. Commonly, the prior is a Gaussian distribution with an identity covariance matrix, and thus $\nabla_{\boldsymbol{\omega}}^2 [-\log p(\boldsymbol{\omega}^*)] = \mathbf{I}$.

The significance of this kernel lies in its relationship with the Fisher information matrix at $\boldsymbol{\omega}^*$ (Immer, 2020; Immer et al., 2021; Kirsch & Gal, 2022), or equivalently, with the linearization of the loss function around $\boldsymbol{\omega}^*$ (Holzmüller et al., 2022). This leads to a Gaussian approximation, which results in a Gaussian predictive posterior distribution when combined with a Gaussian likelihood. The use of the finite-dimensional gradient kernel thus results in an implicit Bayesian linear regression in the context of regression models.

**Posterior Gradient Kernel.** We can use the well-known properties of multivariate normal distributions to marginalize or condition the joint distribution in (4). Following Holzmüller et al. (2022), this allows us to explicitly obtain the posterior gradient kernel given additional $\mathbf{x}_1, \ldots, \mathbf{x}_n$ as:

$$k_{\text{grad} \to \text{post}(\mathbf{x}_1, \ldots, \mathbf{x}_n)}(\mathbf{x}; \mathbf{x}' \mid \boldsymbol{\omega}^*) \tag{8}$$

$$\triangleq \nabla_{\boldsymbol{\omega}} \mu(\mathbf{x}; \boldsymbol{\omega}^*) \left( \sigma_N^{-2} \begin{pmatrix} \nabla_{\boldsymbol{\omega}} \mu(\mathbf{x}_1; \boldsymbol{\omega}^*) \\ \vdots \\ \nabla_{\boldsymbol{\omega}} \mu(\mathbf{x}_n; \boldsymbol{\omega}^*) \end{pmatrix} \begin{pmatrix} \nabla_{\boldsymbol{\omega}} \mu(\mathbf{x}_1; \boldsymbol{\omega}^*) \\ \vdots \\ \nabla_{\boldsymbol{\omega}} \mu(\mathbf{x}_n; \boldsymbol{\omega}^*) \end{pmatrix}^\top + \nabla_{\boldsymbol{\omega}}^2 [-\log p(\boldsymbol{\omega}^*)] \right)^{-1} \nabla_{\boldsymbol{\omega}} \mu(\mathbf{x}'; \boldsymbol{\omega}^*)^\top.$$

The factor $\sigma_N^{-2}$ originates from implicitly conditioning on $Y_i \mid \mathbf{x}_i$, which include observation noise.

Importantly for active learning, the multivariate normal distribution is the maximum entropy distribution for a given covariance matrix, and is thus an upper-bound for the entropy of any distribution with the same covariance matrix. The entropy is given by the log-determinant of the covariance matrix:

$$\mathrm{H}[Y_1, \ldots, Y_n \mid \mathbf{x}_1, \ldots, \mathbf{x}_n] = \frac{1}{2} \log \det(\mathrm{Cov}[\mu(\mathbf{x}_1), \ldots, \mu(\mathbf{x}_n)] + \sigma^2 \mathbf{I}) + C_n, \tag{9}$$

where $C_n \triangleq \frac{n}{2} \log(2\pi e)$ is a constant that only depends on the number of samples $n$. Connecting kernel-based methods to information-theoretic quantities like the expected information gain, we then know that the respective acquisition scores are upper-bounds on the actual expected information gain. Please see §A in the appendix for more details on information-theoretic active learning.

### 3.2   Black-Box Batch Active Learning

We more formally introduce the empirical covariance kernel and compare it to the gradient kernel commonly used for active learning with deep learning models in parameter-space. For non-differentiable models, we show how it can also be derived using a Bayesian model perspective on the hypothesis space.

In addition to being illustrative, this section allows us to connect prediction-based kernels to the kernels used by Holzmüller et al. (2022), which in turns connects them to various SotA active learning methods.

#### 3.2.1   Predictive Covariance Kernel

To perform black-box batch active learning, we directly use the *predictive covariance* of $Y_i|\mathbf{x}_i$ and $Y_j|\mathbf{x}_j$:

$$\mathrm{Cov}_{\boldsymbol{\Omega}}[Y_i; Y_j \mid \mathbf{x}_i, \mathbf{x}_j] = \mathrm{Cov}_{\boldsymbol{\Omega}}[\mu_{\mathbf{x}_i}^{\boldsymbol{\omega}}; \mu_{\mathbf{x}_j}^{\boldsymbol{\omega}}] + \sigma_N^2 \mathbb{1}\{i = j\}. \tag{10}$$

where we have abbreviated $\mu(\mathbf{x}; \boldsymbol{\omega})$ with $\mu_{\mathbf{x}}^{\boldsymbol{\omega}}$, and used the law of total covariance and the fact that the noise is uncorrelated between samples.

> We define the *predictive covariance kernel* $k_{\mathrm{pred}}(\mathbf{x}_i, \mathbf{x}_j)$ as the covariance of the predicted means:
>
> $$k_{\mathrm{pred}}(\mathbf{x}_i; \mathbf{x}_j) \triangleq \mathrm{Cov}_{\boldsymbol{\Omega}}[\mu_{\mathbf{x}_i}^{\boldsymbol{\omega}}; \mu_{\mathbf{x}_j}^{\boldsymbol{\omega}}]. \tag{11}$$
>
> Compared to §3.1, we do not define the covariance via the kernel but the kernel via the covariance.

This is also simply known as the *covariance kernel* in the literature (Shawe-Taylor et al., 2004). We use the prefix *predictive* to make clear that we look at the covariance of the predictions. The resulting Gram matrix is equal the covariance matrix of the predictions and positive definite (for positive $\sigma_N$ and otherwise positive semi-definite) and thus a valid kernel.

#### 3.2.2   Empirical Predictive Covariance Kernel

For $K$ sampled model parameters $\boldsymbol{\omega}_1, \ldots, \boldsymbol{\omega}_K \sim \mathrm{p}(\boldsymbol{\omega})$—for example, the members of a deep ensemble—the *empirical predictive covariance kernel* $k_{\widehat{\mathrm{pred}}}(\mathbf{x}_i; \mathbf{x}_j)$ is the empirical estimate:

$$k_{\widehat{\mathrm{pred}}}(\mathbf{x}_i; \mathbf{x}_j) \triangleq \widehat{\mathrm{Cov}}_{\boldsymbol{\Omega}}[\mu_{\mathbf{x}_i}^{\boldsymbol{\omega}}; \mu_{\mathbf{x}_j}^{\boldsymbol{\omega}}] = \frac{1}{K} \sum_{k=1}^{K} \left( \mu_{\mathbf{x}_i}^{\boldsymbol{\omega}_k} - \frac{1}{K} \sum_{l=1}^{K} \mu_{\mathbf{x}_i}^{\boldsymbol{\omega}_l} \right) \left( \mu_{\mathbf{x}_j}^{\boldsymbol{\omega}_k} - \frac{1}{K} \sum_{l=1}^{K} \mu_{\mathbf{x}_j}^{\boldsymbol{\omega}_l} \right) \tag{12}$$

$$= \left\langle \frac{1}{\sqrt{K}}(\bar{\mu}_{\mathbf{x}_i}^{\boldsymbol{\omega}_1}, \ldots, \bar{\mu}_{\mathbf{x}_i}^{\boldsymbol{\omega}_K}), \frac{1}{\sqrt{K}}(\bar{\mu}_{\mathbf{x}_j}^{\boldsymbol{\omega}_1}, \ldots, \bar{\mu}_{\mathbf{x}_j}^{\boldsymbol{\omega}_K}) \right\rangle, \tag{13}$$

with centered predictions $\bar{\mu}_{\mathbf{x}}^{\boldsymbol{\omega}_k} \triangleq \mu_{\mathbf{x}}^{\boldsymbol{\omega}_k} - \frac{1}{K} \sum_{l=1}^{K} \mu_{\mathbf{x}}^{\boldsymbol{\omega}_l}$. As we can write this kernel as an inner product, it also immediately follows that the empirical predictive covariance kernel is a valid kernel and positive semi-definite.

### 3.2.3 Differentiable Models

Similar to Holzmüller et al. (2022, §C.1), we show that the posterior gradient kernel is a first-order approximation of the (predictive) covariance kernel. This section explicitly conditions on $\mathcal{D}^{\text{train}}$. The result is simple but instructive:

> **Proposition 3.1.** *The* posterior *gradient kernel* $k_{\text{grad} \to \text{post}(\mathcal{D}^{train})}(\mathbf{x}_i; \mathbf{x}_j \mid \boldsymbol{\omega}^*)$ *is an approximation of the predictive covariance kernel* $k_{\text{pred}}(\mathbf{x}_i; \mathbf{x}_j)$.

*Proof.* We use a first-order Taylor expansion of the mean function $\mu(\mathbf{x}; \boldsymbol{\omega})$ around $\boldsymbol{\omega}^*$:

$$\mu(\mathbf{x}; \boldsymbol{\omega}) \approx \mu(\mathbf{x}; \boldsymbol{\omega}^*) + \nabla_{\boldsymbol{\omega}}\mu(\mathbf{x}; \boldsymbol{\omega}^*) \underbrace{(\boldsymbol{\omega} - \boldsymbol{\omega}^*)}_{\triangleq \Delta\boldsymbol{\omega}}. \tag{14}$$

Choose $\boldsymbol{\omega}^* = \mathbb{E}_{\boldsymbol{\omega} \sim p(\boldsymbol{\omega}|\mathcal{D}^{\text{train}})}[\boldsymbol{\omega}]$ (BMA). Then we have $\mathbb{E}_{p(w|\mathcal{D}^{\text{train}})}[\mu(\mathbf{x}; \boldsymbol{\omega})] \approx \mu(\mathbf{x}; \boldsymbol{\omega}^*)$. Overall this yields:

$$k_{\text{pred}}(\mathbf{x}_i; \mathbf{x}_j) = \text{Cov}_{\boldsymbol{\omega} \sim p(\boldsymbol{\omega}|\mathcal{D}^{\text{train}})}[\mu(\mathbf{x}_i; \boldsymbol{\omega}); \mu(\mathbf{x}_j; \boldsymbol{\omega})] \tag{15}$$

$$\approx \mathbb{E}_{\boldsymbol{\omega}^* + \Delta\boldsymbol{\omega} \sim p(w|\mathcal{D}^{\text{train}})}[\langle \nabla_{\boldsymbol{\omega}}\mu_{\mathbf{x}_i}^{\boldsymbol{\omega}^*} \Delta\boldsymbol{\omega}, \nabla_{\boldsymbol{\omega}}\mu_{\mathbf{x}_j}^{\boldsymbol{\omega}^*} \Delta\boldsymbol{\omega}\rangle] \tag{16}$$

$$= \nabla_{\boldsymbol{\omega}}\mu_{\mathbf{x}_i}^{\boldsymbol{\omega}^*} \, \mathbb{E}_{\boldsymbol{\omega}^* + \Delta\boldsymbol{\omega} \sim p(w|\mathcal{D}^{\text{train}})}[\Delta\boldsymbol{\omega}\Delta\boldsymbol{\omega}^\top] \, \nabla_{\boldsymbol{\omega}}\mu_{\mathbf{x}_j}^{\boldsymbol{\omega}^* \top} \tag{17}$$

$$= \nabla_{\boldsymbol{\omega}}\mu(\mathbf{x}_i; \boldsymbol{\omega}^*) \, \text{Cov}[\boldsymbol{\Omega} \mid \mathcal{D}^{\text{train}}] \, \nabla_{\boldsymbol{\omega}}\mu(\mathbf{x}_j; \boldsymbol{\omega}^*)^\top \tag{18}$$

$$\approx k_{\text{grad} \to \text{post}(\mathcal{D}^{\text{train}})}(\mathbf{x}_i; \mathbf{x}_j \mid \boldsymbol{\omega}^*). \tag{19}$$

The intermediate expectation is the model covariance $\text{Cov}[\boldsymbol{\Omega} \mid \mathcal{D}^{\text{train}}]$ as $\boldsymbol{\omega}^*$ is the BMA. For the last step, we use the generalized Gauss-Newton (GGN) approximation (Immer et al., 2021; Kirsch & Gal, 2022) and approximate the inverse of the covariance using the Hessian of the negative log likelihood at $\boldsymbol{\omega}^*$:

$$\text{Cov}[\boldsymbol{\Omega} \mid \mathcal{D}^{\text{train}}]^{-1} \approx \nabla_{\boldsymbol{\omega}}^2[-\log p(\boldsymbol{\omega}^* \mid \mathcal{D}^{\text{train}})] \tag{20}$$

$$= \nabla_{\boldsymbol{\omega}}^2[-\log p(\mathcal{D}^{\text{train}} \mid \boldsymbol{\omega}^*) - \log p(\boldsymbol{\omega}^*)] \tag{21}$$

$$\overset{(GGN)}{\approx} \sigma_N^{-2} \sum_i \nabla_{\boldsymbol{\omega}}\mu(\mathbf{x}^{\text{train}}_i; \boldsymbol{\omega}^*)^\top \nabla_{\boldsymbol{\omega}}\mu(\mathbf{x}^{\text{train}}_i; \boldsymbol{\omega}^*) \tag{22}$$

$$- \nabla_{\boldsymbol{\omega}}^2 \log p(\boldsymbol{\omega}^*), \tag{23}$$

where we have first used Bayes' theorem and that $p(\mathcal{D}^{\text{train}})$ vanishes under differentiation—it is constant in $\boldsymbol{\omega}$. Secondly, the Hessian of the negative log likelihood is just the outer product of the gradients divided by the noise variance in the homoscedastic regression case. $\nabla_{\boldsymbol{\omega}}^2[-\log p(\boldsymbol{\omega}^*)]$ is the prior term. This matches (8). $\square$

### 3.2.4 Non-Differentiable Models

How can we apply the above result to non-differentiable models? In the following, we use a Bayesian view on the hypothesis space to show that we can connect the empirical predictive covariance kernel to a gradient kernel in this case, too. With $\hat{\boldsymbol{\Omega}} \triangleq (\boldsymbol{\omega}_1, \ldots, \boldsymbol{\omega}_K)$ fixed—e.g. these could be the different parameters of the members of an ensemble—we introduce a latent $\Psi$ to represent the index of the 'true' hypothesis $\boldsymbol{\omega}_\psi \in \hat{\boldsymbol{\Omega}}$ from this empirical hypothesis space $\hat{\boldsymbol{\Omega}}$, which we want to identify. This is similar to QbC (Seung et al., 1992). In essence, the latent $\Psi$ takes on the role of $\boldsymbol{\Omega}$ from the previous section, and we are interested in learning the 'true' $\Psi$ from additional data. We, thus, examine the kernels for $\Psi$, as opposed to $\boldsymbol{\Omega}$.

Specifically, we model $\Psi$ using a one-hot categorical distribution, that is a multinomial distribution from which we draw one sample: $\Psi \sim \text{Multinomial}(\boldsymbol{q}, 1)$, with $\boldsymbol{q} \in S^{K-1}$ parameterizing the distribution, where $S^{K-1}$ denotes the $K - 1$ simplex in $\mathbb{R}^K$. Then, $\boldsymbol{q}_k = p(\Psi = e_k)$, where $e_k$ denotes the $k$-th unit vector; and $\sum_{k=1}^K \boldsymbol{q}_k = 1$. For the corresponding predictive mean $\tilde{\mu}(\mathbf{x}; \Psi)$, we have:

$$\tilde{\mu}(\mathbf{x}; \Psi) \triangleq \mu(\mathbf{x}; \boldsymbol{\omega}_\Psi) = \langle \mu(\mathbf{x}; \cdot), \Psi \rangle, \tag{24}$$

where we use $\boldsymbol{\omega}_\psi$ to denote the $\boldsymbol{\omega}_k$ when we have $\psi = e_k$ in slight abuse of notation, and $\mu(\mathbf{x}; \cdot) \in \mathbb{R}^K$ is a column vector of the predictions $\mu(\mathbf{x}; \boldsymbol{\omega}_k)$ for $\mathbf{x}$ for all $\boldsymbol{\omega}_k$. This follows from $\psi$ being a one-hot vector.

We now examine this model and its kernels. The BMA of $\tilde{\mu}(\mathbf{x}; \Psi)$ matches the previous empirical mean if we choose $\boldsymbol{q}$ to have an uninformative[4] uniform distribution over the hypotheses ($\boldsymbol{q}_k \triangleq \frac{1}{K}$):

$$\tilde{\mu}(\mathbf{x}; \boldsymbol{q}) \triangleq \mathbb{E}_{\mathrm{p}(\psi)}[\mu(\mathbf{x}; \boldsymbol{\omega}_\psi)] = \langle \mu(\mathbf{x}; \cdot), \boldsymbol{q} \rangle = \sum_{\psi=1}^{K} \boldsymbol{q}_\psi \mu(\mathbf{x}; \boldsymbol{\omega}_\psi) = \sum_{\psi=1}^{K} \frac{1}{K} \mu(\mathbf{x}; \boldsymbol{\omega}_\psi). \tag{25}$$

What is the predictive covariance kernel of this model? And what is the posterior gradient kernel for $\boldsymbol{q}$?

---

**Proposition 3.2.**

1. *The predictive covariance kernel $k_{\mathrm{pred}, \psi}(\mathbf{x}_i, \mathbf{x}_j)$ for $\hat{\boldsymbol{\Omega}}$ using uniform $\boldsymbol{q}$ is equal to the empirical predictive covariance kernel $k_{\widehat{\mathrm{pred}}}(\mathbf{x}_i; \mathbf{x}_j)$.*

2. *The 'posterior' gradient kernel $k_{\mathrm{grad}, \psi \to \mathrm{post}(\mathcal{D}^{train})}(\mathbf{x}_i; \mathbf{x}_j)$ for $\hat{\boldsymbol{\Omega}}$ in respect to $\Psi$ using uniform $\boldsymbol{q}$ is equal to the empirical predictive covariance kernel $k_{\widehat{\mathrm{pred}}}(\mathbf{x}_i; \mathbf{x}_j)$.*

---

*Proof.* Like for the previous differentiable model, the BMA of the model parameters $\Psi$ is just $\boldsymbol{q}$: $\mathbb{E}\,\Psi = \boldsymbol{q}$. The first statement immediately follows:

$$k_{\mathrm{pred}, \psi}(\mathbf{x}_i; \mathbf{x}_j) = \mathrm{Cov}_\psi[\tilde{\mu}(\mathbf{x}_i; \psi); \tilde{\mu}(\mathbf{x}_j; \psi)] = \mathbb{E}_{\mathrm{p}(\psi)}[\bar{\mu}_{\mathbf{x}_i}^{\boldsymbol{\omega}_\psi} \bar{\mu}_{\mathbf{x}_j}^{\boldsymbol{\omega}_\psi}] = \frac{1}{K} \sum_\psi \bar{\mu}_{\mathbf{x}_i}^{\boldsymbol{\omega}_\psi} \bar{\mu}_{\mathbf{x}_j}^{\boldsymbol{\omega}_\psi} = k_{\widehat{\mathrm{pred}}}(\mathbf{x}_i; \mathbf{x}_j). \tag{26}$$

For the second statement, we will show that we can express the predictive covariance kernel as a linearization around $\Psi$. We can read off a linearization for $\nabla_\psi \tilde{\mu}(\mathbf{x}_i; \psi)$ from the inner product in Equation (25):

$$\nabla_\psi \tilde{\mu}(\mathbf{x}_i; \psi) = \mu(\mathbf{x}; \cdot)^\top, \tag{27}$$

This allows us to write the predictive covariance kernel as a linearization around $\boldsymbol{q}$:

$$k_{\mathrm{pred}, \psi}(\mathbf{x}_i; \mathbf{x}_j) = \mathrm{Cov}_{\psi \sim \mathrm{p}(\psi)}[\tilde{\mu}(\mathbf{x}_i; \psi); \tilde{\mu}(\mathbf{x}_j; \psi)] \tag{28}$$

$$= \mathbb{E}_{\boldsymbol{q} + \Delta\psi \sim \mathrm{p}(\psi)}[\nabla_\psi \tilde{\mu}(\mathbf{x}_i; \psi)\Delta\psi, \nabla_\psi \tilde{\mu}(\mathbf{x}_j; \psi)\Delta\psi] \tag{29}$$

$$= \nabla_\psi \tilde{\mu}(\mathbf{x}_i; \boldsymbol{q})\, \mathrm{Cov}[\Psi]\, \nabla_\psi \tilde{\mu}(\mathbf{x}_i; \boldsymbol{q})^\top \tag{30}$$

$$= k_{\mathrm{grad}, \psi \to \mathrm{post}(\mathcal{D}^{\mathrm{train}})}(\mathbf{x}_i; \mathbf{x}_j). \tag{31}$$

$\square$

The above gradient kernel is only the posterior gradient kernel in the sense that we have sampled $\boldsymbol{\omega}_\psi$ from the non-differentiable model after inference on training data. The samples themselves are drawn uniformly.

The covariance of the multinomial $\Psi$ is: $\mathrm{Cov}[\Psi] = \mathrm{diag}(\boldsymbol{q}) - \boldsymbol{q}\boldsymbol{q}^\top$. Thus, substituting, we can verify that the posterior gradient kernel is indeed equal to the predictive covariance kernel explicitly once more:

$$k_{\mathrm{grad}, \psi \to \mathrm{post}(\mathcal{D}^{\mathrm{train}})}(\mathbf{x}_i; \mathbf{x}_j) = \nabla_\psi \tilde{\mu}(\mathbf{x}_i; \boldsymbol{q})\,(\mathrm{diag}(\boldsymbol{q}) - \boldsymbol{q}\boldsymbol{q}^\top)\,\nabla_\psi \tilde{\mu}(\mathbf{x}_i; \boldsymbol{q})^\top \tag{32}$$

$$= \mu(\mathbf{x}_i; \cdot)^\top \mathrm{diag}(\boldsymbol{q})\,\mu(\mathbf{x}_j; \cdot) - (\mu(\mathbf{x}_i; \cdot)^\top \boldsymbol{q})\,(\boldsymbol{q}^\top \mu(\mathbf{x}_j; \cdot)) \tag{33}$$

$$= \frac{1}{K} \sum_\psi \mu(\mathbf{x}_i; \boldsymbol{\omega}_\psi)\,\mu(\mathbf{x}_j; \boldsymbol{\omega}_\psi)^\top - \left(\frac{1}{K} \sum_\psi \mu(\mathbf{x}_i; \boldsymbol{\omega}_\psi)\right) \left(\frac{1}{K} \sum_\psi \mu(\mathbf{x}_j; \boldsymbol{\omega}_\psi)\right) \tag{34}$$

$$= k_{\widehat{\mathrm{pred}}}(\mathbf{x}_i; \mathbf{x}_j). \tag{35}$$

---

[4]If we had additional information about the $\hat{\boldsymbol{\Omega}}$—for example, if we had validation losses—we could use that to inform $\boldsymbol{q}$.

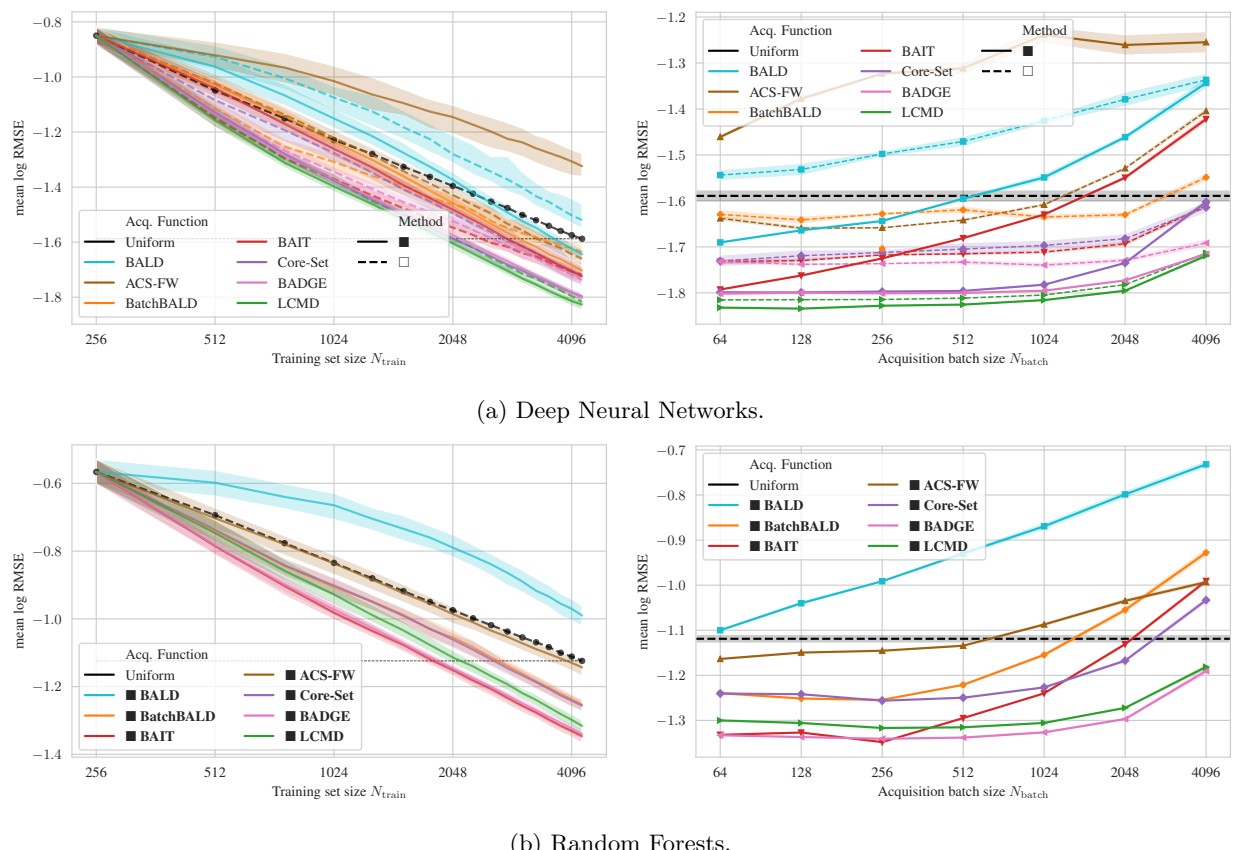

(a) Deep Neural Networks.

(b) Random Forests.

Figure 1: *Mean logarithmic RMSE over 15 regression datasets.* **(a)** For DNNs, we see that black-box ■ methods work as well as white-box □ methods, and in most cases better, with the exception of ACS-FW and BAIT. **(b)** For random forests (100 estimators) with the default hyperparameters from scikit-learn (Pedregosa et al., 2011), we see that black-box methods perform better than the uniform baseline, with the exception of BALD, which uses top-k acquisition. In the appendix, see Table 3 for average performance metrics and Figure 6 and 7 for plots with additional error metrics. Averaged over at least 5 trials.

This demonstrates that a Bayesian model can be constructed on top of a non-differentiable ensemble model. Bayesian inference in this context aims to identify the most suitable member of the ensemble. Given the limited number of samples and likelihood of model misspecification, it is likely that none of the members accurately represents the true model. However, for active learning purposes, the main focus is solely on quantifying the degree of disagreement among the ensemble members.

A similar Bayesian model using Bayesian Model Combination (BMC) could be set up which allows for arbitrary convex mixtures of the ensemble members. This would entail using a Dirichlet distribution $\Psi \sim \text{Dirichlet}(\boldsymbol{\alpha})$ instead of the multinomial distribution. Assuming an uninformative prior ($\boldsymbol{\alpha}_k \triangleq 1/K$), this leads to the same results up to a constant factor of $1 + \sum_k \boldsymbol{\alpha}_k = 2$. This is pleasing because it does not matter whether we use a multinomial or Dirichlet distribution, that is: whether we take a hypothesis space view with a 'true' hypothesis or accept that our model is likely misspecified and we are dealing with a mixture of models, the results are the same up to a constant factor.

**Application to DNNs, BNNs, and Other Models.** The proposed approach has relevance due to its versatility, as it can be applied to a wide range of models that can be consistently queried for prediction, including deep ensembles (Lakshminarayanan et al., 2016), Bayesian neural networks (BNNs) (Blundell et al., 2015; Gal & Ghahramani, 2015), and non-differentiable models. The kernel used in this approach is simple to implement and scales in the number of empirical predictions per sample, rather than in the parameter space, as seen in other methods such as Ash et al. (2021).

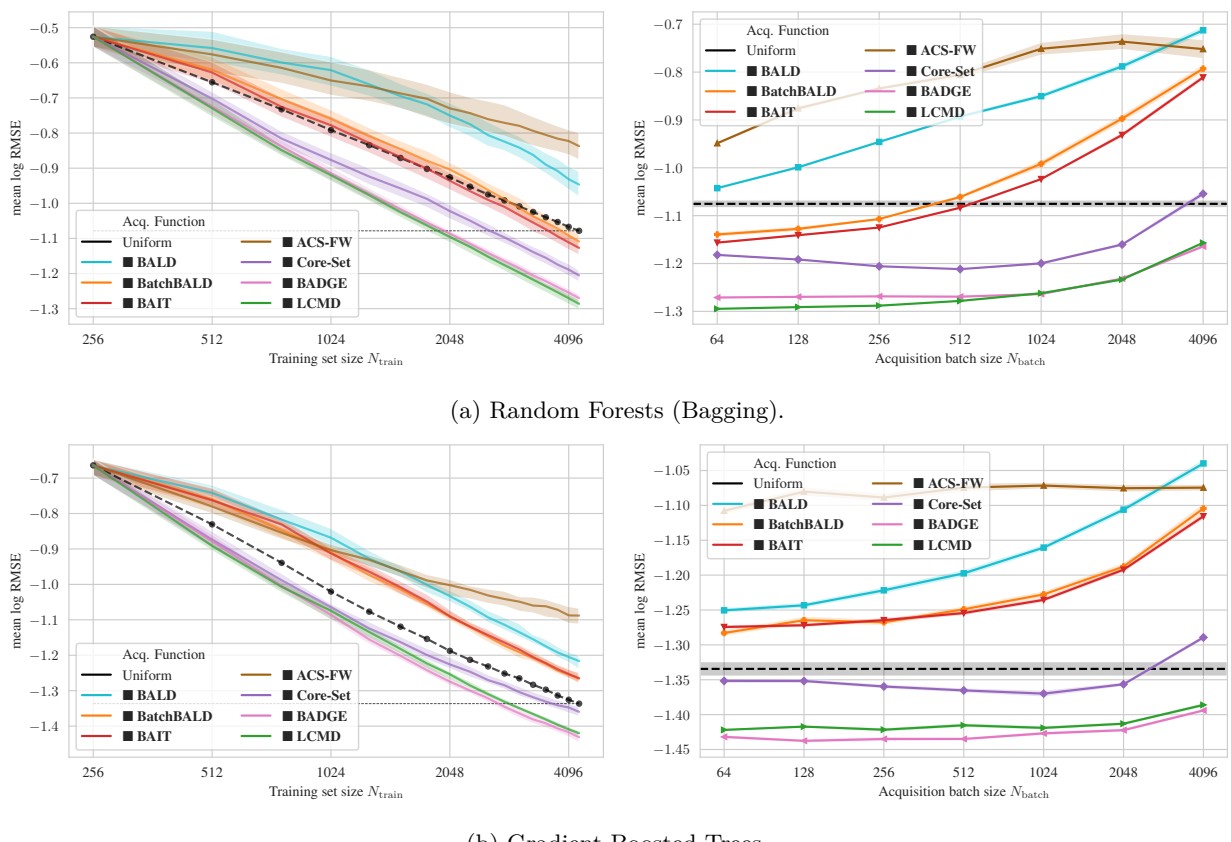

(a) Random Forests (Bagging).

(b) Gradient-Boosted Trees.

Figure 2: *Mean logarithmic RMSE over 15 regression datasets (cont'd).* For random forests using bagging (**a**) with 10 bootstrapped training sets, and for gradient-boosted trees (Dorogush et al., 2018) (**b**) with a *virtual ensemble* of 20 members, we see that only a few of the black-box methods perform better than the uniform baseline: LCMD, BADGE and CoreSet. We hypothesize that the virtual ensembles and a bagged ensemble of random forests do not express as much predictive disagreement which leads to worse performance for active learning. In the appendix, see Table 3 for average performance metrics and Figure 8 and 9 for plots with additional error metrics. Averaged over at least 5 trials.

## 4    Results

We follow the evaluation from Holzmüller et al. (2022) and use their framework to ease comparison. This allows us to directly compare to several SotA methods in a regression setting, respectively their kernel-based analogues. Specifically, we compare to the following popular deep active learning methods: BALD (Houlsby et al., 2011), BatchBALD (Kirsch et al., 2019), BAIT (Ash et al., 2021), BADGE (Ash et al., 2019), ACS-FW (Pinsler et al., 2019), Core-Set (Sener & Savarese, 2017)/FF-Active (Geifman & El-Yaniv, 2017), and LCMD (Holzmüller et al., 2022). The kernels and selection methods used are detailed in Table 1 in the appendix. We also compare to the random selection baseline ('Uniform'). We use 15 large tabular datasets from the UCI Machine Learning Repository (Dua & Graff, 2017) and the OpenML benchmark suite (Vanschoren et al., 2014) for our experiments: *sgemm* (SGEMM GPU kernel performance); *wec_sydney* (Wave Energy Converters); *ct_slices* (Relative location of CT slices on axial axis); *kegg_undir* (KEGG Metabolic Reaction Network - Undirected); *online_video* (Online Video Characteristics and Transcoding Time); *query* (Query Analytics Workloads); *poker* (Poker Hand); *road* (3D Road Network - North Jutland, Denmark); *mlr_knn_rng*; *fried*; *diamonds*; *methane*; *stock* (BNG stock); *protein* (physicochemical-protein); *sarcos* (SARCOS data). See Table 2 in the appendix for more details.

**Experimental Setup.** We use the same experimental setup and hyperparameters as Holzmüller et al. (2022). We report the logarithmic RMSE averaged over 5 trials for each dataset and method. For ensembles,

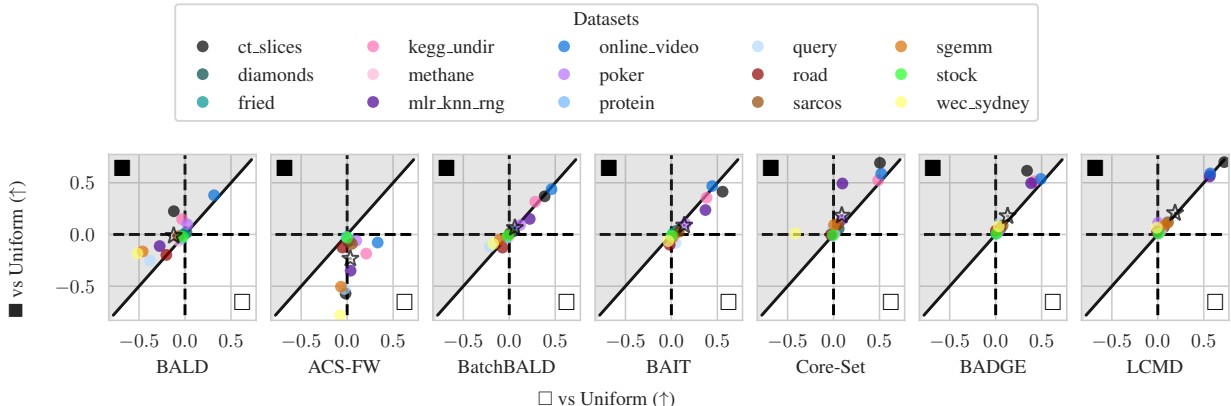

Figure 3: *Average Logarithmic RMSE by regression datasets for DNNs:* ■ *vs* □ *(vs Uniform).* Across acquisition functions, the performance of black-box methods is highly correlated with the performance of white-box methods, even though black-box methods make fewer assumptions about the model. We plot the improvement of the white-box □ method over the uniform baseline on the x-axis (so for datasets with markers right of the dashed vertical lines, the white-box method performs better than uniform) and the improvement of the black-box ■ method over the uniform baseline on the y-axis (so for datasets with markers above the dashed horizontal lines, the black-box method performs better than uniform). For datasets with markers in the ■ *diagonal half*, the black-box method performs better than the white-box method. The average over all datasets is marked with a star ⋆. Surprisingly, on average over all acquisition rounds, the black-box methods perform slightly better than the white-box methods for all but ACS-FW and BAIT. In the appendix, see Figure 5 for the final acquisition round and Table 2 for details on the datasets. Averaged over at least 5 trials.

we compute the performance for each ensemble member separately, enabling a fair comparison to the non-ensemble methods. Performance differences can thus be attributed to the acquisition function, rather than the ensemble. We used A100 GPUs with 40GB of GPU memory.

**Ensemble Size.** For deep learning, we use a small ensemble of 10 models, which is sufficient to achieve good performance. This ensemble size can be considered 'small' in the regression setting (Lázaro-Gredilla & Figueiras-Vidal, 2010; Zhang & Zhou, 2013), whereas in Bayesian Optimal Experiment Design much higher sample counts are regularly used (Foster et al., 2021). In many cases, training an ensemble of regression models is still fast and could be considered cheap compared to the cost of acquiring additional labels. For non-differentiable models, we experiment with: a) random forests (Breiman, 2001), using the different trees as ensemble members; b) bagged random forests, each with 100 trees; and c) a virtual ensemble of gradient-boosted decision trees (Prokhorenkova et al., 2017). For random forests, we use the implementation provided in scikit-learn (Pedregosa et al., 2011) with default hyperparameters, that is using 100 trees per forest. We use the predictions from each tree as a virtual ensemble member. We do not perform any hyperparameter tuning. We also report results for random forests with bagging, where we train a real ensemble of 10 random forests. For gradient-boosted decision trees, we use a virtual ensemble of up to 20 members with early stopping using a validation set[5]. We use the implementation in CatBoost (Dorogush et al., 2018). We do not perform any hyperparameter tuning.

**Black-Box vs White-Box Deep Active Learning.** In Figure 1a and 3, we see that B³AL is competitive with white-box active learning, when using BALD, BatchBALD, BAIT, BADGE, and Core-Set. On average, B³AL outperforms the white-box methods on the 15 datasets we analyzed (excluding ACS-FW and BAIT). We hypothesize that this is due to the implicit Fisher information approximation in the white-box methods (Kirsch & Gal, 2022), which is not as accurate in the low data regime as the more explicit approximation in B³AL via ensembling.

---

[5]If the virtual ensemble creation fails because there are no sufficiently many trees due to early stopping, we halve the ensemble size and retry. This was only needed for the *poker* dataset.

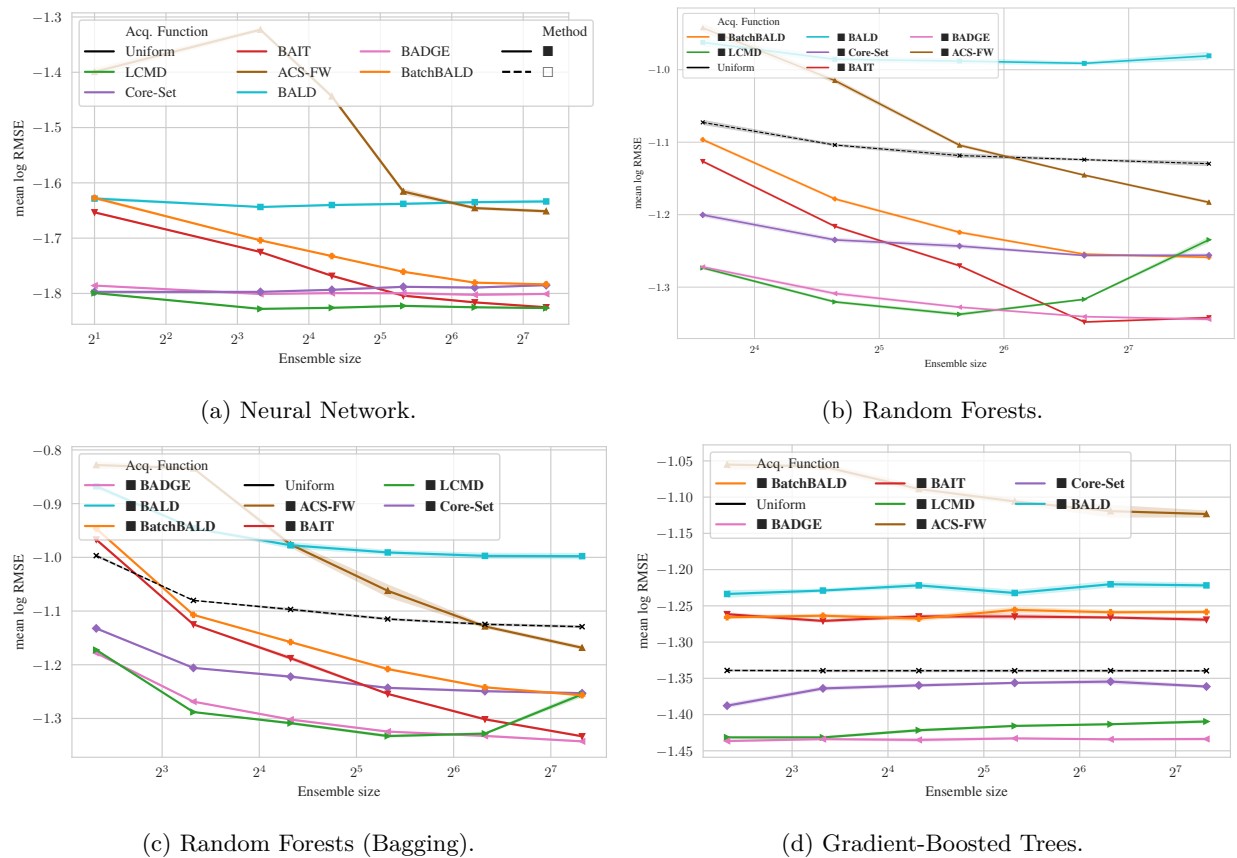

(a) Neural Network.

(b) Random Forests.

(c) Random Forests (Bagging).

(d) Gradient-Boosted Trees.

Figure 4: *Ensemble Size Ablation: Mean logarithmic RMSE over 15 regression datasets.* For neural networks and random forests, increasing the ensemble size generally improves the performance (with the exception of LMCD, for which performance first improves and then degrades). Uniform sampling improves slightly with increasing ensemble size due to better model predictions; compared to the improvements of the active learning methods, these increases are small, however. For gradient-boosted trees with virtual ensembles, increasing the ensemble size (5–160) does not provide a performance boost. Averaged over at least 5 trials.

**Why Can Black-Box Methods Outperform White-Box Methods?** Following §3, white-box and black-box methods are based on kernels, which can be seen as different *approximations* of the predictive covariance kernel. White-box methods implicitly assume that the predictive covariance kernel is well approximated by the Fisher information kernel and the gradient kernel (Kirsch & Gal, 2022). However, Long (2022) demonstrated that this assumption does not always hold, particularly in low data regimes, where a Gaussian might not approximate the parameter distribution well. Instead, Long (2022) suggests using a multimodal distribution. In these situations, methods that employ ensembling, such as B³AL, to approximate the predictive covariance kernel can be more robust. The different ensemble members can reside in different modes of the parameter distribution, allowing black-box methods to outperform their white-box counterparts.

**Non-Differentiable Models.** In Figure 1b, 2a, and 2b, we observe that B³AL is effective for non-differentiable models, including random forests and gradient-boosted decision trees. BALD for non-differentiable models can be considered equivalent to QbC (Seung et al., 1992), while BatchBALD for non-differentiable models can be viewed as equivalent to QbC with batch acquisition (Nguyen et al., 2012). For random forests, all methods except BALD (using top-k selection) outperform uniform acquisition. However, for random forests with bagging and gradient-boosted decision trees, B³AL surpasses random acquisition only when employing LMCD and BADGE. This may be attributed to the reduced disagreement within a virtual ensemble for gradient-boosted decision trees and between distinct random forests. In particular, random forests with bagging appear to support this explanation, as a single random forest seems to exhibit

more disagreement among its individual trees than an ensemble of random forests with bagging does between different forests. This is evident in the superior overall active learning performance of the single random forest compared to the ensemble of random forests with bagging, c.f. Figure 1b and 2a.

## 4.1 Ablations

Here, we discuss two ablations of B³AL: the effect of the ensemble size and the effect of the acquisition batch sizes.

**Acquisition Batch Size.** The right-hand plots in Figures 1 and 2 show that the performance generally decreases with increasing acquisition batch size for all acquisition approaches. Comparing white-box and black-box methods in Figure 1a, we see that only at the largest batch size of 4096, the white-box methods perform as well as the black-box methods for neural networks.

**Ensemble Batch Size.** Figure 4 shows the effect of the ensemble size on the performance of B³AL for random forests. Increasing the ensemble size generally improves the performance for all acquisition approaches except LMCD, for which performance first improves and then degrades. The performance increase with uniform sampling is due to better model predictions. This shows that the improvements stemming from more informative sample acquisition by the active learning methods due to increasing ensemble sizes can be significantly larger than the improvements due to better model predictions.

**Best-performing Kernels and Selection Methods.** Holzmüller et al. (2022) evaluate different kernels and variants of selection methods based on the SotA methods. The best-performing kernels and selection methods are shown in Table 5 in the appendix. We run an ablation comparing black-box methods with the best-performing white-box kernels from Holzmüller et al. (2022). These variants do not actually match prior art—we use the same names for the acquisition functions to denote the origin of the variants. The results are shown in Appendix B.1. We observe that B³AL also performs on par or better than the best-performing white-box kernels.

## 5 Conclusion and Future Work

In this paper, we have demonstrated the effectiveness of a simple extension to kernel-based methods that utilizes empirical predictions rather than gradient kernels. This modification enables black-box batch active learning with good performance. Importantly, B³AL also generalizes to non-differentiable models, an area that has received limited attention as of late.

The main limitation of our proposed approach lies in the acquisition of a sufficient amount of empirical predictions. This could be a challenge, particularly when using deep ensembles with larger models or non-differentiable models that cannot be parallelized efficiently. Our experiments using virtual ensembles indicate that the diversity of the ensemble members plays a crucial role in determining the performance. The main limitation of the empirical comparisons is that we only consider regression tasks. Extending the results to classification is an important direction for future work.

Overall, our results partially answer one of the research questions posed by Kirsch & Gal (2022): how do prediction-based methods compare to parameter-based ones? We find that for regression the prediction-based methods are competitive with the parameter-based methods in batch active learning.

### Acknowledgements

The author would like to thank the anonymous TMLR reviewers for their patience and kind feedback during the review process. Their feedback has significantly improved this work. Likewise, many thanks to David Holzmüller for their constructive and helpful feedback, as well as for making their framework and results easily available. AK is supported by the UK EPSRC CDT in Autonomous Intelligent Machines and Systems (grant reference EP/L015897/1). ChatGPT was used to suggest specific text and table edits.

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

# A    Information-Theoretic Active Learning

The following section provides a brief overview of information-theoretic active learning, focusing on the expected information gain/BALD and greedy batch selection. Other information-theoretic quantities can be examined similarly. We refer to Kirsch & Gal (2022) for more details. For other selection methods, we refer to Holzmüller et al. (2022).

**Expected Information Gain (EIG)** (Lindley, 1956) is based on the mutual information. For a Bayesian model, the EIG between the parameters $\boldsymbol{\Omega}$ and the predictions $Y \mid x$ is defined as:

$$\mathrm{I}[\boldsymbol{\Omega}; Y \mid \mathbf{x}] = \mathrm{H}[\boldsymbol{\Omega}] - \mathrm{H}[\boldsymbol{\Omega} \mid Y, \mathbf{x}] = \int \mathrm{p}(\boldsymbol{\omega}, y \mid \mathbf{x}) \log \frac{\mathrm{p}(\boldsymbol{\omega} \mid y, \mathbf{x})}{\mathrm{p}(\boldsymbol{\omega})} \, d\boldsymbol{\omega} \, dy, \tag{36}$$

where $\mathrm{I}[\boldsymbol{\Omega}; Y \mid \mathbf{x}]$ is the mutual information between the parameters $\boldsymbol{\Omega}$ and the model predictions $Y$ given the data $\mathbf{x}$; $\mathrm{H}[\boldsymbol{\Omega}]$ is the entropy of the parameters $\boldsymbol{\Omega}$; and $\mathrm{H}[\boldsymbol{\Omega} \mid Y, \mathbf{x}]$ is the conditional entropy of the parameters $\boldsymbol{\Omega}$ in expectation over the predictions $\boldsymbol{\Omega}$ and the model predictions $Y \mid \mathbf{x}$. The EIG measures the amount of information that can be gained about the parameters $\boldsymbol{\Omega}$ by observing the predictions $Y \mid \mathbf{x}$ on average, where we think of the entropy $\mathrm{H}[\boldsymbol{\Omega}]$ as capturing the uncertainty about the parameters $\boldsymbol{\Omega}$.

Computing the EIG can be difficult when focused on the model parameters, as we need to compute the conditional entropy $\mathrm{H}[\boldsymbol{\Omega} \mid Y, \mathbf{x}]$—yet this is what many recent approaches effectively attempt to do via the Fisher information (Kirsch & Gal, 2022). (We do not need to compute the entropy of the parameters $\mathrm{H}[\boldsymbol{\Omega}]$, as it does not depend on the data $\mathbf{x}$ and can be dropped from the optimization objective.)

**BALD** (Houlsby et al., 2011) examines the expected information gain in *prediction space* instead of parameter space, using the symmetry of the mutual information:

$$\mathrm{I}[\boldsymbol{\Omega}; Y \mid \mathbf{x}] = \mathrm{I}[Y; \boldsymbol{\Omega} \mid \mathbf{x}] = \mathrm{H}[Y \mid \mathbf{x}] - \mathrm{H}[Y \mid \boldsymbol{\Omega}, \mathbf{x}] = \int \mathrm{p}(\boldsymbol{\omega}, y \mid \mathbf{x}) \log \frac{\mathrm{p}(y \mid \mathbf{x}, \boldsymbol{\omega})}{\mathrm{p}(y \mid \mathbf{x})} \, d\boldsymbol{\omega} \, dy, \tag{37}$$

This can be much easier to evaluate by sampling from $\boldsymbol{\Omega}$ without the need for additional Bayesian inference. BatchBALD (Kirsch et al., 2019) is an extension of BALD to the batch acquisition of samples $\{\mathbf{x}_i^{\mathrm{acq}}\}$ for classification tasks:

$$\mathrm{I}[\boldsymbol{\Omega}; Y_1, \ldots, Y_B \mid \mathbf{x}_1, \ldots, \mathbf{x}_B] = \mathrm{H}[Y_1, \ldots, Y_B \mid \mathbf{x}_1, \ldots, \mathbf{x}_B] - \sum_i \mathrm{H}[Y_i \mid \mathbf{x}_i, \boldsymbol{\Omega}], \tag{38}$$

using the mutual information between the parameters $\boldsymbol{\omega}$ and the predictions $\{Y_i\}$ on an acquisition candidate batch $\{\mathbf{x}_i\}$, and thus takes the correlations between samples into account.

This joint entropy can be upper-bounded by the corresponding entropy of multivariate normal distribution with the same covariance matrix, while the conditional entropy term is precisely the entropy of a normal distribution given our model assumptions for the observation noise, yielding an upper-bound overall.

The same approach can be applied to other information quantities like the expected predicted information gain (Bickford Smith et al., 2023). Unlike Kirsch & Gal (2022), which examines the connection between Fisher information (weight-space) and information-theoretic quantities, this exposition shows the connection between predictions and information-theoretic quantities.

**Greedy Acquisition** is a heuristic to find the subset of size $B$ in $\mathcal{D}^{\mathrm{pool}}$ that maximizes $\mathcal{A}(\{\mathbf{x}_i\}, \mathrm{p}(\boldsymbol{\omega}))$. Finding the optimal solution is NP-hard (Schrijver et al., 2003). Kirsch et al. (2019) find that $\mathcal{A}_{\mathrm{BatchBALD}}$ is submodular and that iteratively adding the sample $\mathbf{x}^{\mathrm{acq}}{}_i$ with the highest $\mathcal{A}_{\mathrm{BatchBALD}}(\mathbf{x}^{\mathrm{acq}}{}_1, \ldots, \mathbf{x}^{\mathrm{acq}}{}_i, \mathrm{p}(\boldsymbol{\omega}))$ to the acquisition batch size is a $1 - \mathit{1}/_e$-approximation. This is a well-known result in submodular optimization and is known as the *greedy algorithm* (Nemhauser et al., 1978). For the entropy approximation, finding an optimal solution is equivalent to finding the mode of the k-DPP (Kulesza & Taskar, 2011).

# B  Results

Table 1: Kernels and selection methods for SotA methods. Taken from Holzmüller et al. (2022). See Holzmüller et al. (2022) for more details.

| Known as | Selection method | Kernel | Remark |
|---|---|---|---|
| BALD (Houlsby et al., 2011) | MaxDiag | $k_{\text{ll}\to\mathcal{X}_{\text{train}}}$ | for GP with kernel $k$ |
| BatchBALD (Kirsch et al., 2019) | MaxDet-P | $k_{\text{ll}\to\mathcal{X}_{\text{train}}}$ | for GP with kernel $k$, proposed for classification |
| BAIT (Ash et al., 2021) | Bait-FB-P | $k_{\text{ll}\to\mathcal{X}_{\text{train}}}$ | |
| ACS-FW (Pinsler et al., 2019) | FrankWolfe-P | $k_{\text{ll}\to\text{acs-rf-hyper}(512)}$ | |
| Core-Set (Sener & Savarese, 2017), FF-Active (Geifman & El-Yaniv, 2017) | MaxDist-TP* | $k_{\text{ll}}$ | proposed for classification |
| BADGE (Ash et al., 2019) | KMeansPP-P | $k_{\text{ll}\to\mathcal{X}_{\text{train}}}$ | |
| LCMD | LCMD-TP | $k_{\text{grad}\to\text{sketch}(512)}$ | |

\* This refers to their simpler k-center-greedy selection method.

Table 2: *Overview over used data sets.* See Ballester-Ripoll et al. (2019); Neshat et al. (2018); Graf et al. (2011); Shannon et al. (2003); Deneke et al. (2014); Anagnostopoulos et al. (2018); Savva et al. (2018); Friedman (1991); Ślęzak et al. (2018). The second column entries are hyperlinks to the respective web pages. Taken from Holzmüller et al. (2022).

| Short name | Initial pool set size | Test set size | Number of features | Source | OpenML ID | Full name |
|---|---|---|---|---|---|---|
| sgemm | 192000 | 48320 | 14 | UCI | | SGEMM GPU kernel performance |
| wec_sydney | 56320 | 14400 | 48 | UCI | | Wave Energy Converters |
| ct_slices | 41520 | 10700 | 379 | UCI | | Relative location of CT slices on axial axis |
| kegg_undir | 50407 | 12921 | 27 | UCI | | KEGG Metabolic Reaction Network (Undirected) |
| online_video | 53748 | 13756 | 26 | UCI | | Online Video Characteristics and Transcoding Time |
| query | 158720 | 40000 | 4 | UCI | | Query Analytics Workloads |
| poker | 198720 | 300000 | 95 | UCI | | Poker Hand |
| road | 198720 | 234874 | 2 | UCI | | 3D Road Network (North Jutland, Denmark) |
| mlr_knn_rng | 88123 | 22350 | 132 | OpenML | 42454 | mlr_knn_rng |
| fried | 31335 | 8153 | 10 | OpenML | 564 | fried |
| diamonds | 41872 | 10788 | 29 | OpenML | 42225 | diamonds |
| methane | 198720 | 300000 | 33 | OpenML | 42701 | Methane |
| stock | 45960 | 11809 | 9 | OpenML | 1200 | BNG(stock) |
| protein | 35304 | 9146 | 9 | OpenML | 42903 | physicochemical-protein |
| sarcos | 34308 | 8896 | 21 | GPML | | SARCOS data |

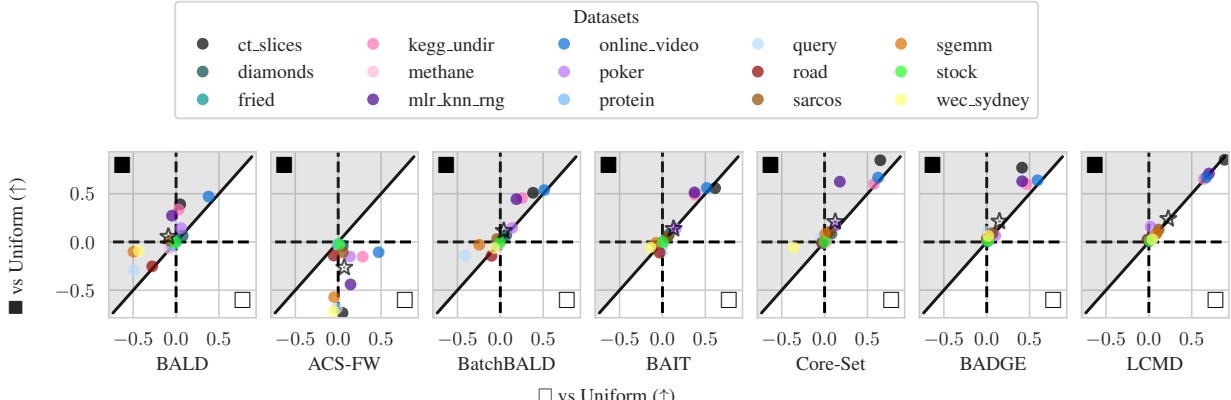

Figure 5: *Final Logarithmic RMSE by regression datasets for DNNs: ■ vs □ (vs Uniform).* Across acquisition functions, the performance of black-box methods is highly correlated with the performance of white-box methods, even though black-box methods make fewer assumptions about the model. We plot the improvement of the white-box method (□) over the uniform baseline on the x-axis, so for datasets with markers left of the dashed vertical lines, the white-box method performs better than uniform, and the improvement of the black-box method (■) over the uniform baseline on the y-axis, so for datasets with markers above the dashed horizontal lines, the black-box method performs better than uniform. Similarly, for datasets with markers in the ■ region, the black-box method performs better than the white-box method. The average over all datasets is marked with a star ⋆.

Table 3: *Average performance of black-box ■ and white-box □ batch active learning acquisition functions using DNNs.* On average, for five acquisition methods, the black-box method performs better than the white-box method. Cf. Figure 3, which analyzes the final epoch.

| Acquisition function | MAE | RMSE | 95% | 99% | MAXE |
|---|---|---|---|---|---|
| Uniform | -1.935 | -1.401 | -0.766 | -0.163 | 1.107 |
| ■ **BALD** | -1.794 | -1.389 | -0.713 | -0.221 | 0.946 |
| □ BALD | -1.717 | -1.278 | -0.611 | -0.070 | 1.090 |
| ■ **BatchBALD** | -1.865 | -1.465 | -0.792 | -0.303 | 0.892 |
| □ BatchBALD | -1.894 | -1.462 | -0.808 | -0.287 | 0.912 |
| □ BAIT | -1.999 | -1.540 | -0.895 | -0.356 | 0.895 |
| ■ **BAIT** | -1.892 | -1.489 | -0.817 | -0.328 | 0.881 |
| □ ACS-FW | -1.935 | -1.436 | -0.793 | -0.222 | 1.031 |
| ■ **ACS-FW** | -1.678 | -1.168 | -0.509 | 0.085 | 1.278 |
| ■ **Core-Set** | -1.988 | -1.585 | -0.926 | -0.435 | **0.831** |
| □ Core-Set | -1.924 | -1.491 | -0.831 | -0.308 | 0.929 |
| ■ **BADGE** | -2.042 | -1.579 | -0.931 | -0.383 | 0.948 |
| □ BADGE | -2.009 | -1.531 | -0.897 | -0.331 | 1.009 |
| ■ **LCMD** | **-2.048** | **-1.609** | **-0.965** | **-0.437** | 0.874 |
| □ LCMD | -2.033 | -1.589 | -0.940 | -0.401 | 0.913 |

Table 4: *Average performance of black-box* ■ *batch active learning acquisition functions on non-differentiable models.*

(a) Random Forests

| Acquisition function | MAE | RMSE | 95% | 99% | MAXE |
|---|---|---|---|---|---|
| Uniform | -1.516 | -0.977 | -0.292 | 0.290 | 1.380 |
| ■ **BALD** | -1.222 | -0.815 | -0.106 | 0.365 | 1.348 |
| ■ **BatchBALD** | -1.449 | -1.070 | -0.401 | 0.064 | **1.195** |
| ■ **BAIT** | -1.582 | **-1.158** | **-0.485** | **0.021** | 1.198 |
| ■ **ACS-FW** | -1.502 | -0.989 | -0.310 | 0.266 | 1.379 |
| ■ **Core-Set** | -1.449 | -1.071 | -0.403 | 0.059 | 1.196 |
| ■ **BADGE** | **-1.618** | -1.150 | -0.480 | 0.070 | 1.273 |
| ■ **LCMD** | -1.564 | -1.116 | -0.450 | 0.097 | 1.270 |

(b) Random Forests (Bagging)

| Acquisition function | MAE | RMSE | 95% | 99% | MAXE |
|---|---|---|---|---|---|
| Uniform | -1.438 | -0.931 | -0.237 | 0.317 | 1.375 |
| ■ **BALD** | -1.154 | -0.771 | -0.070 | 0.386 | 1.349 |
| ■ **BatchBALD** | -1.283 | -0.923 | -0.245 | 0.205 | 1.263 |
| ■ **BAIT** | -1.304 | -0.944 | -0.269 | 0.184 | 1.254 |
| ■ **ACS-FW** | -1.206 | -0.732 | -0.019 | 0.503 | 1.471 |
| ■ **Core-Set** | -1.369 | -1.030 | -0.377 | **0.071** | **1.194** |
| ■ **BADGE** | **-1.524** | -1.089 | -0.413 | 0.114 | 1.261 |
| ■ **LCMD** | -1.496 | **-1.099** | **-0.433** | 0.074 | 1.219 |

(c) Gradient-Boosted Decision Trees

| Acquisition function | MAE | RMSE | 95% | 99% | MAXE |
|---|---|---|---|---|---|
| Uniform | -1.691 | -1.179 | -0.543 | 0.065 | 1.224 |
| ■ **BALD** | -1.445 | -1.044 | -0.379 | 0.143 | 1.217 |
| ■ **BatchBALD** | -1.484 | -1.092 | -0.435 | 0.080 | 1.167 |
| ■ **BAIT** | -1.485 | -1.088 | -0.433 | 0.095 | 1.191 |
| ■ **ACS-FW** | -1.498 | -0.999 | -0.340 | 0.246 | 1.326 |
| ■ **Core-Set** | -1.600 | -1.215 | -0.566 | -0.053 | **1.086** |
| ■ **BADGE** | **-1.728** | **-1.264** | **-0.629** | -0.048 | 1.157 |
| ■ **LCMD** | -1.689 | -1.253 | -0.613 | **-0.055** | 1.132 |

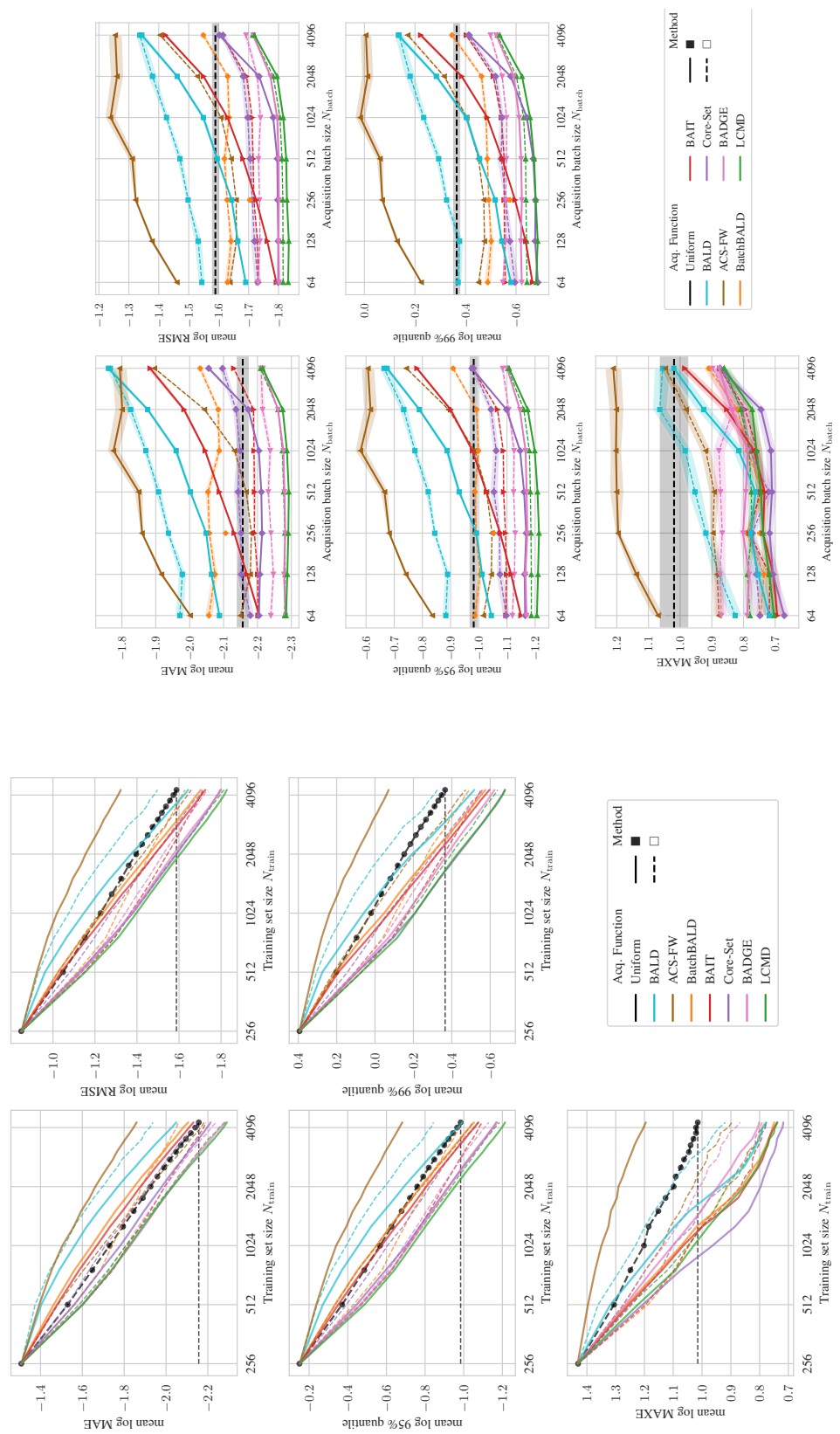

(a) Learning Curves

(b) Acquisition Batch Sizes

Figure 6: *DNNs: Error Metrics over 15 regression datasets.* We report mean absolute error (MAE), root mean squared error (RMSE), 95% and 99% quantiles, and the maximum error (MAXE). Averaged over 5 trials.

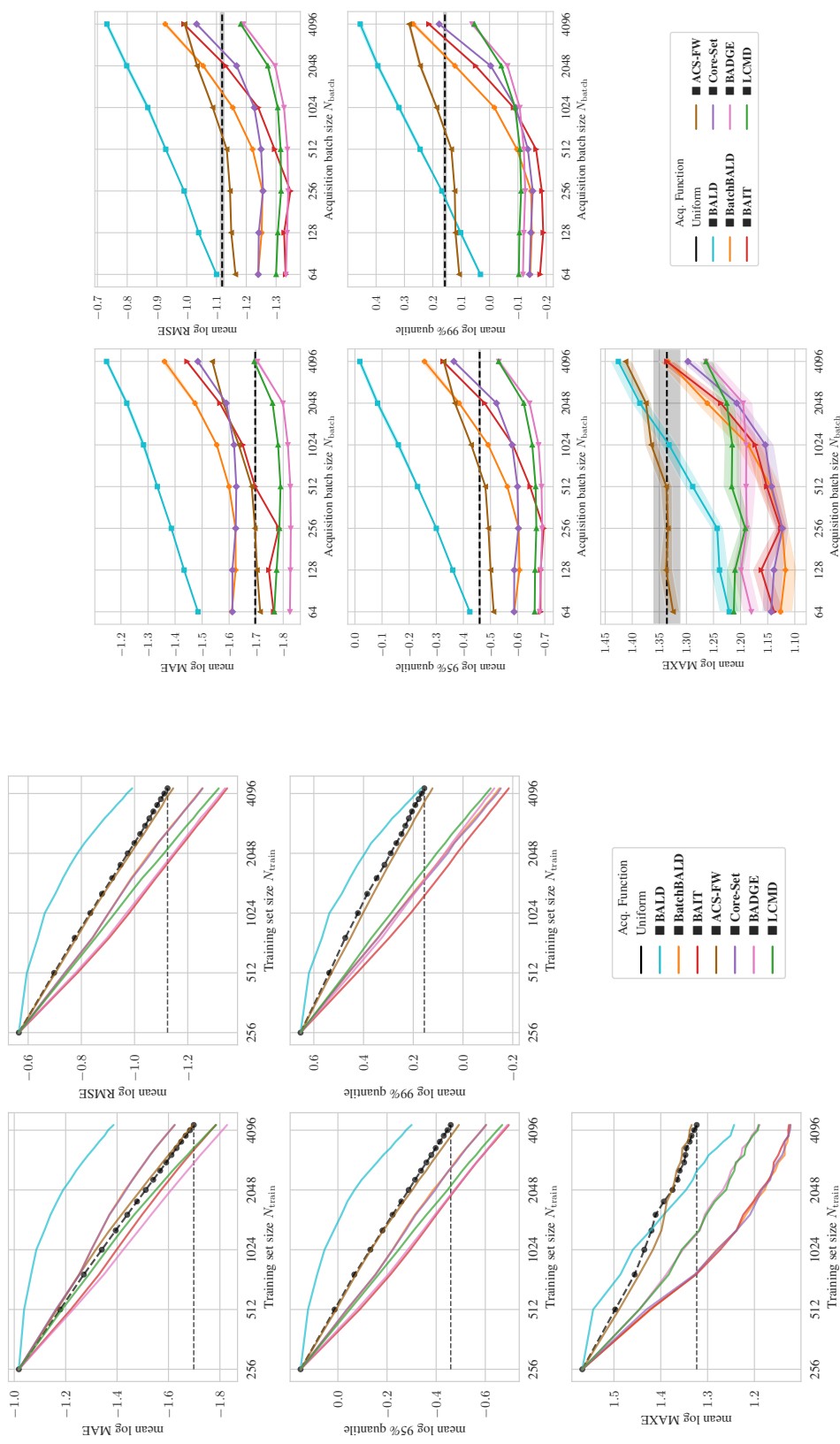

(a) Learning Curves

(b) Acquisition Batch Size

Figure 7: *Random Forests: Error Metrics over 15 regression datasets (cont'd)*. We report mean absolute error (MAE), root mean squared error (RMSE), 95% and 99% quantiles, and the maximum error (MAXE). Averaged over 5 trials.

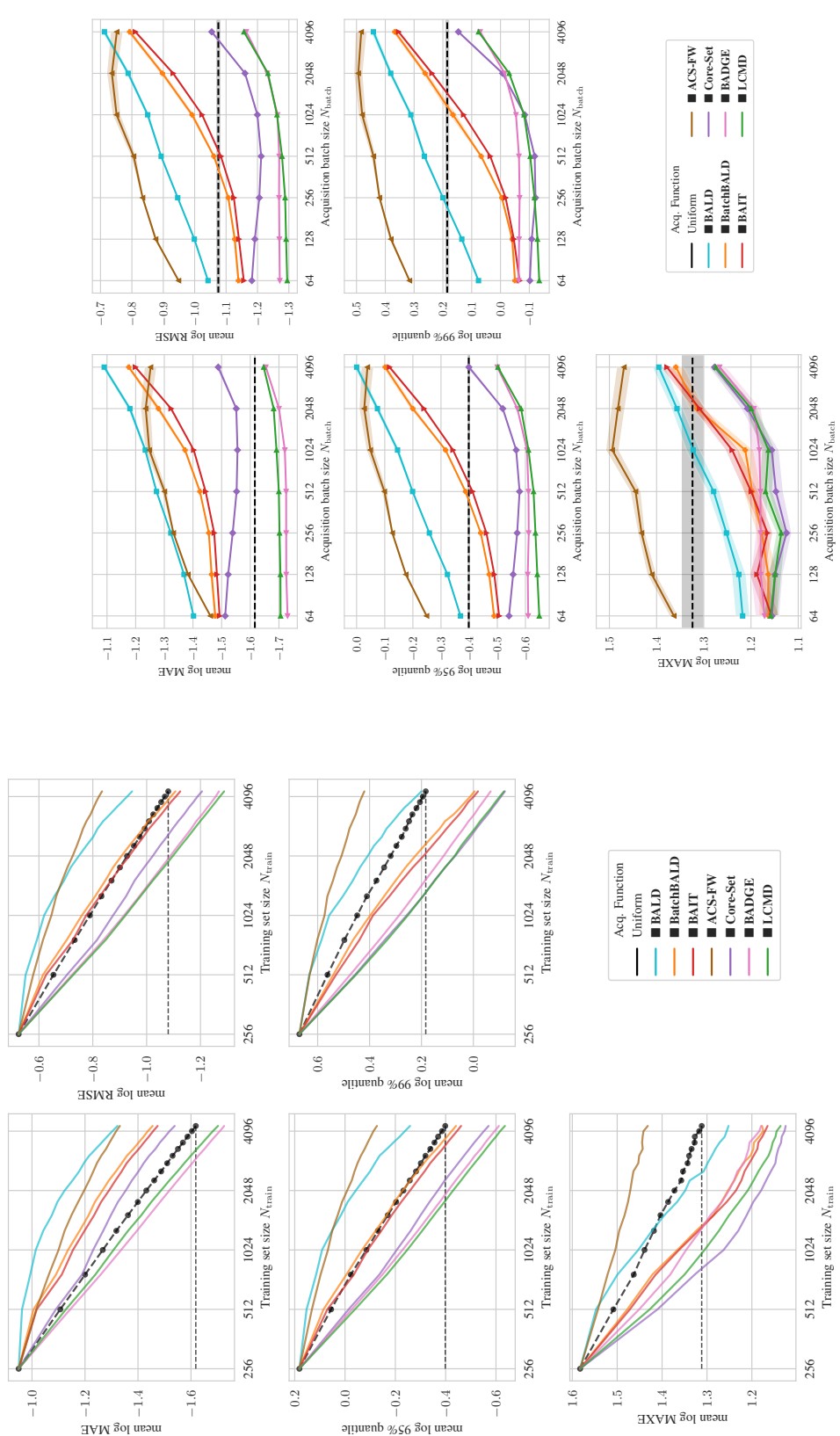

(a) Learning Curves

(b) Acquisition Batch Size

Figure 8: *Random Forests (Bagging): Error Metrics over 15 regression datasets (cont'd).* We report mean absolute error (MAE), root mean squared error (RMSE), 95% and 99% quantiles, and the maximum error (MAXE). Averaged over 5 trials.

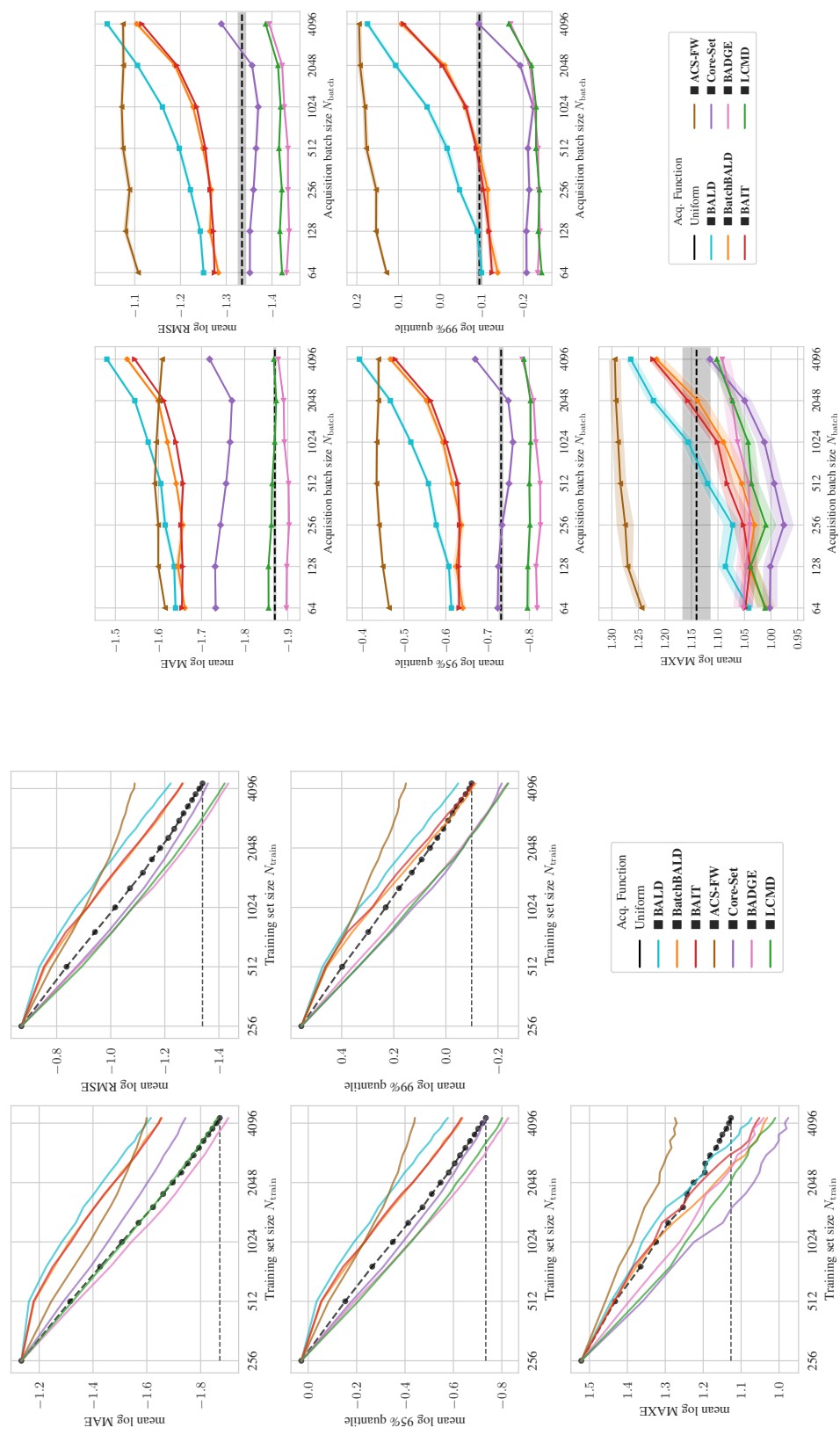

(a) Learning Curves

(b) Acquisition Batch Size

Figure 9: *Gradient-Boosted Trees with Virtual Ensemble: Error Metrics over 15 regression datasets (cont'd).* We report mean absolute error (MAE), root mean squared error (RMSE), 95% and 99% quantiles, and the maximum error (MAXE). Averaged over 5 trials.

## B.1 Comparison on Neural Networks with Best-Performing Kernels and Selection Methods

Table 5: Best-performing white-box kernels and selection methods based on SotA methods according to results from Holzmüller et al. (2022). Taken from Holzmüller et al. (2022).

| Based on | Selection method | Kernel |
|---|---|---|
| BALD (Houlsby et al., 2011) | MaxDiag | $k_{\text{grad}\to\text{sketch}(512)\to\text{acs-rf}(512)}$ |
| BatchBALD (Kirsch et al., 2019) | MaxDet-P | $k_{\text{grad}\to\text{sketch}(512)\to\mathcal{X}_{\text{train}}}$ |
| BAIT (Ash et al., 2021) | Bait-F-P | $k_{\text{grad}\to\text{sketch}(512)\to\mathcal{X}_{\text{train}}}$ |
| ACS-FW (Pinsler et al., 2019) | FrankWolfe-P | $k_{\text{grad}\to\text{sketch}(512)\to\text{acs-rf-hyper}(512)}$ |
| Core-Set (Sener & Savarese, 2017), FF-Active (Geifman & El-Yaniv, 2017) | MaxDist-P | $k_{\text{grad}\to\text{sketch}(512)\to\mathcal{X}_{\text{train}}}$ |
| BADGE (Ash et al., 2019) | KMeansPP-P | $k_{\text{grad}\to\text{sketch}(512)\to\text{acs}-\text{rf}(512)}$ |
| LCMD | LCMD-TP | $k_{\text{grad}\to\text{sketch}(512)}$ |

Table 6: *Average performance of black-box* ■ *and white-box* □ *batch active learning acquisition functions using DNNs for Table 5.* On average, for five acquisition methods, the black-box method performs better than the white-box method. Cf. Figure 11, which analyzes the final epoch.

| Acquisition function | MAE | RMSE | 95% | 99% | MAXE |
|---|---|---|---|---|---|
| Uniform | -1.935 | -1.401 | -0.766 | -0.163 | 1.107 |
| ■ **BALD** | -1.794 | -1.389 | -0.713 | -0.221 | 0.946 |
| □ BALD | -1.779 | -1.372 | -0.692 | -0.191 | 0.979 |
| □ BatchBALD | -1.915 | -1.512 | -0.844 | -0.349 | 0.866 |
| ■ **BatchBALD** | -1.865 | -1.465 | -0.792 | -0.303 | 0.892 |
| □ BAIT | -2.014 | -1.586 | -0.928 | -0.413 | 0.864 |
| ■ **BAIT** | -1.892 | -1.489 | -0.817 | -0.328 | 0.881 |
| □ ACS-FW | -1.979 | -1.544 | -0.892 | -0.363 | 0.920 |
| ■ **ACS-FW** | -1.678 | -1.168 | -0.509 | 0.085 | 1.278 |
| ■ **Core-Set** | -1.988 | -1.585 | -0.926 | -0.435 | **0.831** |
| □ Core-Set | -1.916 | -1.515 | -0.845 | -0.351 | 0.865 |
| ■ **BADGE** | -2.042 | -1.579 | -0.931 | -0.383 | 0.948 |
| □ BADGE | -2.009 | -1.570 | -0.915 | -0.387 | 0.921 |
| ■ **LCMD** | **-2.048** | **-1.609** | **-0.965** | **-0.437** | 0.874 |
| □ LCMD | -2.033 | -1.589 | -0.940 | -0.401 | 0.913 |

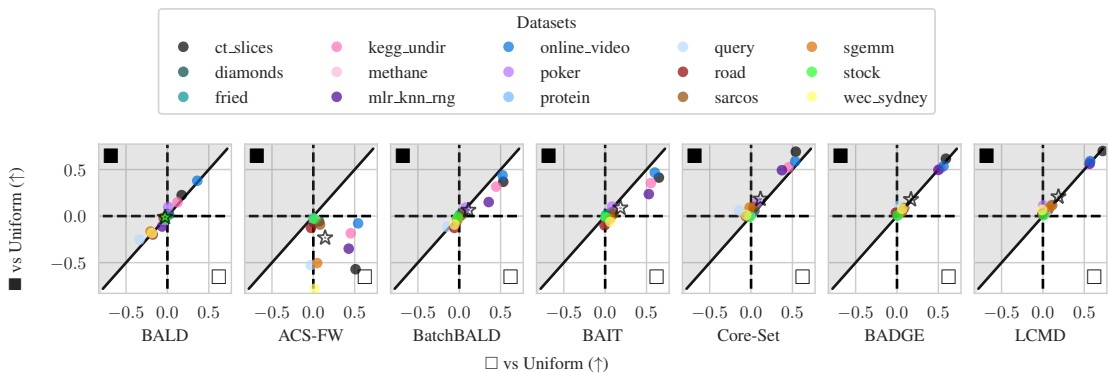

Figure 10: *Average Logarithmic RMSE by regression datasets for DNNs: ■ vs □ (vs Uniform).*

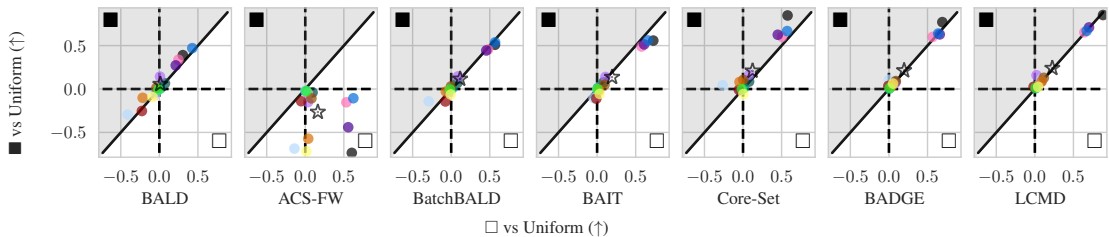

Figure 11: *Final Logarithmic RMSE by regression datasets for DNNs: ■ vs □ (vs Uniform).*

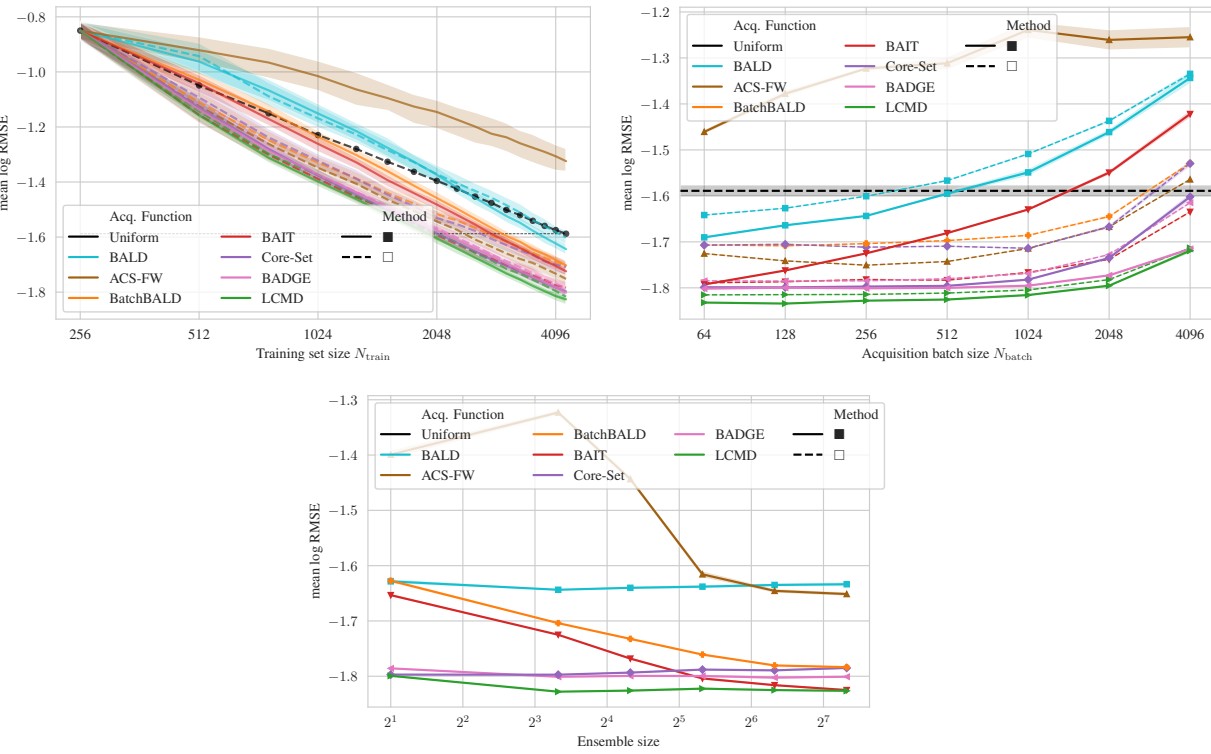

Figure 12: *Mean logarithmic RMSE over 15 regression datasets for Deep Neural Networks using the best-performing white-box kernels and selection methods from Table 5.* Even for the best-performing white-box kernels and selection methods as determined by Holzmüller et al. (2022), the black-box approach outperforms the white-box approach overall.

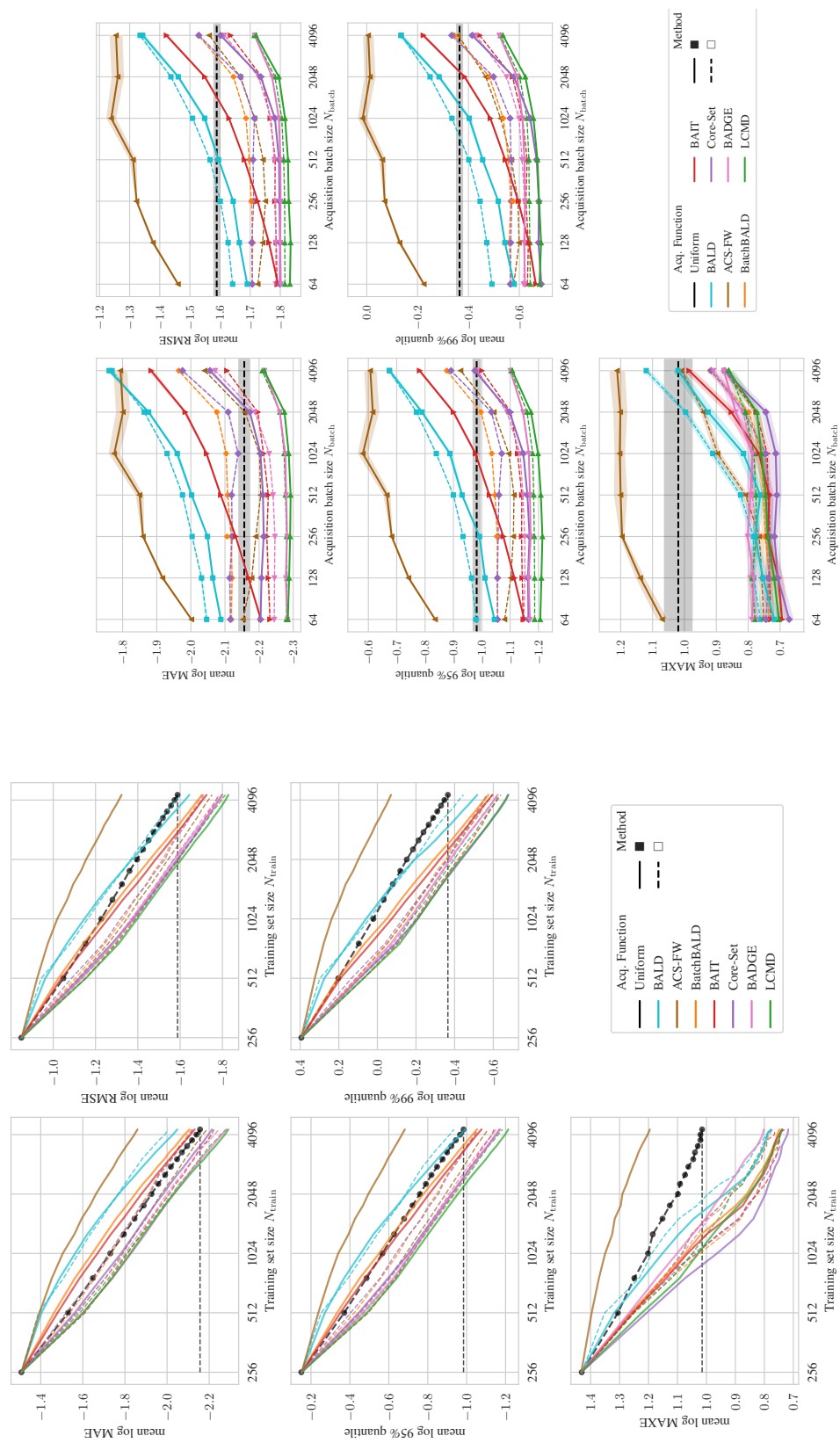

(a) Learning Curves

(b) Acquisition Batch Sizes

Figure 13: *DNNs: Error Metrics over 15 regression datasets using the strongest white-box kernels from Table 5.* We report mean absolute error (MAE), root mean squared error (RMSE), 95% and 99% quantiles, and the maximum error (MAXE). Averaged over 5 trials.

