# OpenReview forum: "Black-Box Batch Active Learning for Regression"
_TMLR — Accepted by TMLR_

### Review · Reviewer_h9yU · 2023-02-25

**Summary Of Contributions:**

This paper proposes black-box active learning for regression task. Existing works on white box active learning can only be applied to white-box models. This paper addresses this limitation by proposing black box methods which is applicable to black-box models. Experimental results showed that the proposed black-box methods can outperform existing white-box methods.

**Audience:**

Yes

**Claims And Evidence:**

Yes

**Requested Changes:**

Please clarify my concerns in the above Weaknesses

**Strengths And Weaknesses:**

Strengths:
1. Clear motivation about why black-box active learning can be useful
2. Detailed explanation about related background and the method
3. Experimental results look good.

Weaknesses:
1. It's not clear why this paper focuses on regression tasks. Classification tasks are also very popular (if not more popular) than regression tasks.
2. It's not clear why the (proposed) black-box methods can outperform white-box methods. Intuitively, white-box methods can access more information (model internals, such as architecture, or weights), thus better suited to the specific models. I am not sure why the black-box methods can outperform white-box methods as shown in Figure 1.

---

> ### Author Response · Authors · 2023-03-21
> **Thanks for your review!**
>
> We want to thank the reviewer for their feedback and for acknowledging the motivation behind our black-box active learning approach. We appreciate their recognition of the detailed explanation of our method and background and their satisfaction with our experimental results.
>
> We have revised the paper in light of their feedback to address the concerns they raised:
>
> **Reason for Focusing on Regression Tasks**
>
> The primary objective of our paper is to compare equivalent white and black-box active learning methods. The study by Holzmueller et al., which our paper builds upon, presents a unified and principled way to evaluate various active learning methods for regression tasks. This approach facilitates a straightforward comparison of white and black-box methods on an equal footing.
>
> Therefore, we chose to concentrate on regression tasks. Applying the same approach to classification tasks would necessitate using Laplace approximations, Monte Carlo sampling, or expectation propagation, complicating a fair comparison. Translating existing classification methods for active learning is not a straightforward process.
>
> **How Black-box Methods Can Outperform White-box Methods**
>
> White-box and black-box methods are based on kernels, which can be seen as different approximations of the predictive covariance kernel. White-box methods implicitly assume that the predictive covariance kernel is well approximated by the Fisher information kernel and the gradient kernel (Kirsch & Gal, 2022). However, Long (2022) demonstrated that this assumption does not always hold, particularly in low data regimes, where a Gaussian does not approximate the parameter distribution well. Instead, Long (2022) suggests using a multimodal distribution.
>
> In these situations, methods that employ ensembling, such as B3AL, to approximate the predictive covariance kernel are more robust. The different ensemble members can reside in different modes of the parameter distribution, allowing black-box methods to outperform their white-box counterparts.
>
> Please let us know if this helps address your questions.
>
> Apologies for the delay!
>
> Best wishes,\
> the Authors

---

### Review · Reviewer_pr7j · 2023-03-06

**Summary Of Contributions:**

This paper has proposed a batch active learning method for black-box (potentially non-differnetiable) models in regression tasks. The proposed method is based on a recent development of the kernel-based perspective of active learning and introduces prediction-based kernels, and hence can be used to modify various existing active learning algorithms for the black-box setting. Empirical evaluations verify that black-box methods perform comparably with or even surpasses their white-box counterparts.

**Audience:**

Yes

**Broader Impact Concerns:**

I do not have any concern on the ethical implications of this work.

**Claims And Evidence:**

Yes

**Requested Changes:**

A small suggestion in addition to the points listed under Weaknesses above:
- One line above equation (1): when can $\Omega$ be bootstrapped training data? I find this section on page 3 much easier to understand if you just say that $\Omega$ represent the model parameters, so you can perhaps just say that $\Omega$ is the model parameters, and then say (perhaps in a footnote) that in some special cases, it can also be others as well such as bootstrapped training data.

**Strengths And Weaknesses:**

Strengths:
- The proposed method based on prediction-based kernels is nicely motivated and positioned with respect to the previous works.
- The method, although simple, indeed leads to well-performing black-box active learning methods which are empirically shown to be competitive with their corresponding white-box methods. So the proposed technique can be a useful contribution to the field of black-box active learning and a nice baseline for future works in this area.

Weaknesses:
- I think the authors should discuss why the predictive covariance kernel (and also the empirical one) is a positive semi-definite kernel. The posterior gradient kernel from previous works is PSD since it's an inner product in some feature space, but the predictive covariance kernel proposed in this work isn't clear to me.
- The writing of the paper is clear overall, but there are some places in the technical sections which can use some more polishing and better explanation, to make it easier to understand for users not familiar with the field. For example, first line of Section 3.2.2., are $\omega_1,\ldots,\omega_K$ the model parameters of $K$ models from an ensemble? More importantly, I find the explanation in Section 3.2.4 not easy to understand, so perhaps this section can be made clearer. In equation (28), are you saying that in this case, every member of the ensemble is a model resulting from BMA using equation (28)? What's the dimension of $\mu(x;\cdot)$ in equation (27)? Does the symbol $\psi$ represent an index from $1,\ldots,K$ (as in equation 28) or a vector (as in equation 27)? Overall, I suggest that in Section 3.2.4, whenever you explain a term (e.g., equation 28), you can immediate point out what does this term correspond to in Sections 3.2.1-3.2.3, this makes it easier for readers to see the big picture.
- Can add error bars in Figure 1? With error bars, it's difficult to see whether the differences are significant.
- I find Figure 2 difficult to read. For example, initially I thought the black square in the top left corner refers to the area in of the top left quadrants, which leads to inconsistent interpretations with those from the main text. It took me quite a while to figure out that it actually refers to the entire area above the y=x line. The caption says that on average, the black-box methods perform better the the white-box methods since the stars line in the top left area, but I think that all stars lie pretty close to the y=x line. But I understand that this also shows the strengths of the proposed method since it allows black-box methods to perform on par with the white-box ones. So, I think some modifications are needed for the presentation of Figure 2 and its caption.
- Fourth last line on page 1: "such" -> "such as"

---

### Review · Reviewer_7KgK · 2023-03-06

**Summary Of Contributions:**

The paper introduces a kernel for batch active learning for regression which uses model predictions instead of model parameters (/gradients of model parameters/similar). This allows the method to be applied to non-differentiable models too, like random forests, and means the method no longer scales with number of parameters/final-layer features. The authors show how their method is related to the previous white-box approaches, and compare performance on the benchmark from Holzmuller et al. (2022), where the black-box method usually out-performs the white-box version, and where there is correlation between performance.

**Audience:**

Yes

**Claims And Evidence:**

Yes

**Requested Changes:**

Please see Weaknesses section for the more major requested changes. Especially Weakness 2 and the experiments section.

List of minor changes / typos:
- 2nd line of Related Work: use \citet{} instead of \citep{}
- The authors can give a one-line summary of acquisition function at the beginning of the Related Works section itself instead of later (such as 'it measures the informativeness of...' on page 3).
- page 5 second paragraph: 'paramters'
- page 5 just before Sec 3.2: 'examind'
- Equation 18: 'sim' vs '\sim'
- I'm not sure the prior was defined as p(\psi) before Equation 29
- Page 8: 'Above gradient kernel...' -> 'The above gradient kernel...'
- Figure 2 caption: 'markers left of the dashed vertical lines' -> 'right of the...'
- More important: Figures 1b and 1c are very difficult to read: the font size should be bigger. Also, in general in Figure 1, there are too many overlapping lines, and presentation could be improved.

**Strengths And Weaknesses:**

Strengths

1. I thought the paper was overall well-written and easy to read. I liked the structure / way concepts were built up.
2. The idea itself is simple (look at prediction-space instead of model embeddings or derivatives). The simplicity is a good thing. (Disclaimer: I am assuming people have not tried this method / similar methods before in this field.)
3. The link to non-differentiable models is neat and a strength of the paper, regardless of how raw performance of the models compare to neural networks.

--------------------------

Weaknesses

1. Despite what is said in Strength 2 above, Proposition 3.1 is not new or unexpected, with similar arguments used in the past in non-active-learning works, where the Gaussian approximation / Laplace approximation / linearisation around w* (+ Gauss-Newton approximation) leads to the simple relationship between prediction-space and parameter-space. And very similar arguments, as cited in this paper, were used in Holzmuller et al. (2022). I therefore do not view Prop 3.1 as particularly surprising, rather just a nice property of this method that exists. (This is not a weakness as such, more a comment on how interesting this part is.)

2. Overall I found the experiments section severely lacking: the experiments are enough to show some evidence for why the proposed method might be good, but does not do enough to explore properties of the method. More details / suggestions:
(a) There are no baselines (aside from uniform sampling) in the non-differentiable models experiments. Why is this? If no suitable baselines exist at all in the literature (I am not aware of any, but I am not an expert in active learning), then perhaps some visualisation (or more explanation) of the intuition provided ('the virtual ensemble might not capture disagreement well enough') would be helpful. I found this section (Experiments with non-differentiable models) very short.
(b) The experiments show B3AL generally outperforming the white-box methods, but I still do not really know much about its properties. Figure 2 is nice (showing a correlation between performance), but much more can be done. For example, how does performance change as number of ensembles changes? The authors plot only the mean performance, but how high is the standard deviation across different random seeds (as presumably the set of ensembles / disagreement between ensembles changes)? What is the computation cost of B3AL (with 10 ensembles) compared to the white-box versions?
(c) I notice in Figure 2 that most method/dataset combinations have performance very similar to uniform sampling, with only a few datasets per method performing better than uniform. Figure 1a seems to be hiding this information by taking the mean performance. I think this should be commented on in the paper at least.

---

> ### Author Response · Authors · 2023-03-21
> **Thank you!**
>
> We sincerely appreciate the detailed feedback and are glad you found our paper well-written and liked the structure and sequence of concepts. We are also pleased that you recognized the simplicity of the idea and the potential breadth of applications as strengths of our paper.
>
> Based on your feedback, we have improved the paper, addressing the minor issues you identified. Further to your review:
>
> **Proposition 3.1**
>
> We concur that the propositions are not novel or unexpected. Proposition 3.1 is presented merely to offer instructive framing around the proposed method and to argue that both approaches (black-box and white-box) are approximations of the predictive covariance kernel.
>
> **Reason for Focusing on Regression Tasks (shared with reviewer h9yU)**
>
> The primary aim of our paper is to compare equivalent white and black-box active learning methods. The study by Holzmueller et al., which our paper builds upon, provides a unified and principled way to evaluate various active learning methods for regression tasks. This approach enables a straightforward comparison of white and black-box methods on equal terms.
>
> Hence, we focused on regression tasks. Applying the same approach to classification tasks would require Laplace approximations, Monte Carlo sampling, or expectation propagation, complicating a fair comparison. Adapting existing classification methods for active learning is not a straightforward task.
>
> **Experiment Section**
>
> Firstly, we want to emphasize that the experiments report results on 15 different datasets and four approaches (DNN white box, DNN black box, RF black box (& bagging RF black box), and GBDT black box). We can include per-dataset metrics plots in the camera-ready version's appendix.
>
> However, we agree that more ablations are necessary. We have conducted additional experiments: We have added acquisition batch size ablations for the various approaches (unfortunately, these experiments have not yet been completed). We also concur with the reviewer about adding ensemble size ablations for the camera-ready version.
>
> We propose to run an ensemble in conjunction with the white-box approach for the camera-ready version, which should benefit the white-box approaches and offer insightful information for readers.
>
> In the current revision, we have included results as far as available. Some experiments are still ongoing and need to be completed. We apologize for the less polished appearance due to the first author's illness.
>
> **Baselines for Non-Differentiable Methods**
>
> We have clarified in the experiment section that "BALD" and "BatchBALD" for random forests can be considered equivalent to the old baseline QbC (Seung 1992) and the batch QbC method (Nguyen, 2012). Thus, the non-differentiable results include two baselines from previous research. We mentioned both in the related work section, but reiterating this in the experiment section will make the information more prominent for readers.
>
> ---
>
> For the camera-ready (next revision), we plan to polish the plots (the sizes do not match), finish the current experiment runs, and add an ensemble size ablation.
>
> We apologize for the delay in responding. We hope the update is to your liking.
>
> Thank you so much for your review, and please let us know what you think,\
> the Authors

---

### Review · Reviewer_hT9S · 2023-03-07

**Summary Of Contributions:**

This work investigates the problem of batch active learning for regression problems where the user has black-box access to a neural network (I assume this means that the model parameters, and therefore related information like gradients are unknown, and only predicted regression outputs are known). A Bayesian prediction kernel based on multinomial sampling is presented and it is shown that this kernel can approximate the covariance of predictions for two input states. This observation, along with a recent framework for active learning from given a kernel definition, defines a black-box active learning method. Experimental results show that the proposed method can nearly match white-box active learning for deep learning across several different baseline methods, and also that the proposed method is effective for non-differentiable models like random forests and GBMs.

**Audience:**

Yes

**Broader Impact Concerns:**

No section on broader impacts is included, although this does not significantly affect my assessment.

**Claims And Evidence:**

Yes

**Requested Changes:**

There should be more efforts to compare the proposed method against existing active learning techniques in a black-box setting (possibly with minor modifications). It would also be helpful to discuss which existing methods can be applied or easily adapted to a black-box setting and which necessarily must be applied in the white-box setting (in which case this work provides a novel implementation method).

**Strengths And Weaknesses:**

STRENGTHS:
* The scope of the proposed method is broader than many existing methods because it is effective in non-differentiable settings.
* The proposed method is mathematically sound, the derivation and presentation are clear, and the formulation fits in nicely with recent work that provides a unified perspective for active learning.
* The relatively minor drop in performance for the proposed method when compared to white-box methods are an encouraging demonstration that the proposed method is effective and useful.

WEAKNESSES:

* The term "black-box" is used in different ways in the literature and the meaning in this paper is not fully specified. I assume that "black-box" as used in this paper means that the model regression predictions are available, but not model parameters or gradients. Is that right? If black-box means model gradients are inaccessible and the parameters themselves accessible, it seems like MC-dropout and related techniques could be considered black-box. Clearly defining "black-box" is essential for the reader to understand the context.
* Not enough efforts were devoted to presenting other black-box active learning techniques, or at least basic adaptions of existing techniques into a black-box setting. For example, the Core Set method could be used with an auxiliary network (Inception, VAE, etc.) to encode embedding distances even if the target model parameters are unknown to obtain the intermediate embedding. The results would be more compelling with comparison to other black-box baselines, instead of comparing the proposed black-box results only to white-box results. Similarly, for random forests models, it might be helpful to investigate an MC-dropout type method that looks at the predictions of each tree in the forest as a baseline for comparison (even though this would be white-box) to show performance for non-differentiable white-box vs. non-differentiable black-box.
* The decision to investigate regression rather than classification and regression seems somewhat arbitrary and was not strongly motivated.

---

> ### Author Response · Authors · 2023-03-22
> **Thank you!**
>
> Thank you for your feedback! We are pleased that you consider our method sound and the presentation clear and that you find this avenue of exploration valuable.
>
> We have revised the paper according to your suggestions:
>
> **Clarification of "Black Box"**
>
> We have clarified that we define black-box methods as only using model predictions. Therefore, BALD and BatchBALD are also considered black-box methods, as they only use model predictions for different model draws (using MC Dropout). This clarification has been added to the abstract and the introduction. For our random forest experiments, however, we do utilize the predictions of the random forest's trees. Additionally, we have included an evaluation using a random forest ensemble (via bagging), although its active learning performance is not as strong. (This could be explained by the ensemble capturing less disagreement due to averaging over the disagreement in its internal decision trees).
>
> **Reason for Focusing on Regression Tasks**
>
> Our paper primarily aims to compare equivalent white and black-box active learning methods. The study by Holzmueller et al., upon which our paper builds, presents a unified and principled way to evaluate various active learning methods for regression tasks. This approach directly compares white and black-box methods on equal terms.
>
> As a result, we chose to focus on regression tasks. Applying the same methodology to classification tasks would require using Laplace approximations, Monte Carlo sampling, or expectation propagation, which would complicate a fair comparison. Translating existing classification methods for active learning is not a straightforward process.
>
> **Other Black-Box Baselines**
>
> In this paper, for DNNs, we aim to compare black-box and white-box approaches as fairly as possible. It is worth noting that we already compare seven state-of-the-art active learning techniques across 15 datasets in our analysis.
>
> The baseline you suggest, which involves taking a core-set method and using a different model to provide embeddings or a distance metric, is an exciting idea for selecting a proxy model. However, we prefer not to include it in this work, as it does not directly contribute to the direct comparability. The described method could be considered white box because it uses embeddings. Still, if the model is an auto-encoder, it could also be regarded as black box because the embeddings are the model predictions (similar to ICA).
>
> We believe that using the term "proxy model" might be more appropriate in this case, and it would be intriguing to evaluate active learning for non-differentiable models using proxy models as an extension of "Selection via Proxy: Efficient Data Selection for Deep Learning" by Coleman et al. (2020).
>
> We thank you for your suggestions and hope the changes we have made using your feedback are to your liking.
>
> Apologies for the delays due to the first author's illness, and please let us know what you think,\
> the Authors

---

### Author Response · Authors · 2023-04-18
**Apologies for Delay of Ablation Results**

Dear reviewers and dear AC,

We hope you are well! (We are not sure if the reviewers can see this.)

Some ablations are still running, and we apologize for the delay. More importantly, the PhD thesis submission deadline of the first author is this Friday, which has derailed every other work in the last weeks. Our sincere apologies for this.

Please let us know if and what else we can provide to address your questions and help your decision.

Thank you so much, and best wishes,\
 the Authors

---

> ### Comment · Action_Editors · 2023-04-23
> **Revision**
>
> Dear Authors,
>
> No problem. Please add new experimental results in your revision when they are ready.
>
> AC.

---

### Author Response · Authors · 2023-05-15
**Paper Revision**

Dear reviewers and AE,

We have finally pushed the paper revision. An online diff between the PDFs can be found here: https://draftable.com/compare/YeoHDUjZDgay

We note that we want to rerun the baseline acquisition batch for 256 for VE-CAT for the camera-ready as there is a blip in the acquisition batch size ablation. We also want to provide ensemble size ablations for the other methods. We had to stop experiments due to NeurIPS.

Thank you for being so patient, and please let us know of any further requests. We are looking forward to your additional feedback.

Best wishes,\
the Authors

---

### Decision · Action_Editors · 2023-05-21

**Recommendation:** Accept as is

**Comment:**

The paper was reviewed by four reviewers. They find the paper well-written, and the proposed method novel, interesting, and simple. They raised some concerns about presentation and experiments, as well as limitation to regression setting. The authors have successfully addressed all their concerns in the rebuttals. In the end, all the reviewers are positive about the paper.

I would like to clarify the timeline a little bit. On April 18, the authors reported that they were still running some ablation studies. On May 15, the authors uploaded the revised version of the paper. The delay was caused by some personal issues of the first author, which are completely understandable. I appreciate the efforts in the revision.

**Audience:**

This paper is about a black-box method for batch active learning. It should be interesting to the machine learning community in general.

**Claims And Evidence:**

This paper studies the problem of active learning in the regression setting, where the new training examples are acquired in batches. The paper proposes a black-box kernel-based approach that does not require knowledge about model parameters and gradient, so the proposed method is applicable to non-differentiable models.

The effectiveness of the proposed method was demonstrated by various experiments. The authors also added thorough ablation studies to the revised version of the paper.